# Computationally efficient parameter estimation for high-dimensional ocean biogeochemical models

Skyler Kern[1], Mary E. McGuinn[1], Katherine M. Smith[2], Nadia Pinardi[3], Kyle E. Niemeyer[4], Nicole S. Lovenduski[5], and Peter E. Hamlington[1]

[1]Paul M. Rady Department of Mechanical Engineering, University of Colorado, Boulder, CO, USA
[2]Los Alamos National Laboratory, Los Alamos, NM, USA
[3]Department of Physics and Astronomy, University of Bologna, Bologna, IT
[4]School of Mechanical, Industrial, and Manufacturing Engineering, Oregon State University, Corvallis, OR, USA
[5]Department of Atmospheric and Oceanic Sciences and Institute of Arctic and Alpine Research, University of Colorado, Boulder, CO, USA

**Correspondence:** S. Kern (Skyler.Kern@colorado.edu)

**Abstract.** Biogeochemical (BGC) models are widely used in ocean simulations for a range of applications, but typically include parameters that are determined based on a combination of empiricism and convention. Here, we describe and demonstrate an optimization-based parameter estimation method for high-dimensional (in parameter space) BGC ocean models. Our computationally efficient method combines the respective benefits of global and local optimization techniques and enables simultaneous parameter estimation at multiple ocean locations using multiple state variables. We demonstrate the method for a 17-state-variable BGC model with 51 uncertain parameters, where a one-dimensional (in space) physical model is used to represent vertical mixing. We perform a twin-simulation experiment to test the accuracy of the method in recovering known parameters. We then use the method to simultaneously match multi-variable observational data collected at sites in the subtropical North Atlantic and Pacific. We examine the effects of different objective functions, sometimes referred to as cost functions, which quantify the disagreement between model and observational data. We further examine increasing levels of data sparsity and the choice of state variables used during the optimization. We end with a discussion of how the method can be applied to other BGC models, ocean locations, and mixing representations.

## 1 Introduction

Biogeochemical (BGC) models used in global and regional ocean simulations often contain tens to hundreds of uncertain parameters (e.g., Long et al., 2021). Typically, these parameters are determined by a combination of laboratory experiments, empiricism, expert opinion, and model tuning constraints, with the ultimate goal of achieving good agreement between simulation results and observational data across a range of ocean conditions and locations (Doney et al., 2009). The high computational cost of solving the coupled physical and BGC equations has slowed progress in BGC parameter estimation, yet accurate model parameters are crucial for quantifying key climate processes, such as the strength of the ocean's biological carbon pump (e.g., Henson et al., 2015).

While not the focus of the present study, we face other challenges when attempting to calibration BGC model parameters including limitations on available data and parameter dependencies. Due to the size of the ocean, the vast range of relevant temporal and spatial scales, and complexity of the marine ecosystem, in situ observations taken tend to be sparse and often do not include all quantities present in BGC models. In some situations there is a drastic difference in the number of observed quantities and model state variables. With limited data it can be difficult to constrain parameters related to the state values not observed. The issue of data sparsity can increase the difficulty of handling parameter dependence. The problem is data often will not constrain the individual processes the BGC model is parameterizing. Instead, observations are for particular quantities of interest. For example, observing concentrations of phytoplankton chlorophyll alone could make it difficult to accurately estimate growth and mortality rates since both can be changed (decreased and increased, respectively) to try to produce a similar effect on the different plankton populations. Similarly, both increased growth rates and lower remineralization rates could have the same effect on nutrient concentrations.

In the present study, we address the challenge of calibrating a large set of parameters for a coupled biophysical model by describing and demonstrating a computationally efficient ocean BGC parameter estimation method that takes into account multiple sites and multiple variables. We perform an initial global search of the model parameter space to determine appropriate starting points for subsequent gradient-based local optimizations. The parameter values giving the best locally optimized solution are then taken as the final parameters. We demonstrate the approach by simultaneously optimizing 51 uncertain parameters in a 17-state-variable BGC model at sites in the subtropical North Atlantic and Pacific. To calibrate the BGC model, we couple it to a one-dimensional (1D) vertical ocean mixing model and match several observational fields at each site. We verify the accuracy of the method using a twin simulation experiment (TSE), where we estimate known model parameters from synthetic data generated by a reference model simulation. Subsequent to verification using the TSE, we use the method to estimate parameters for the two sites individually and together using real-world observational data.

The present study extends prior efforts to use optimization methods in BGC model parameter estimation. Matear (1995) used conjugate gradient and simulated annealing methods to calibrate three-, four-, and seven-parameter BGC models, with simulated annealing proving to be more reliable for higher dimensional (in parameter space) models. Oliver et al. (2022) used TSEs to test two derivative-free optimization techniques by attempting to recover six parameters in a nine-component BGC model coupled to a three-dimensional (3D) mixing representation using the transport matrix method (Prieß et al., 2013; Kriest et al., 2017; Kriest, 2017; Sauerland et al., 2019). Of the six parameters, five were recovered and the model results were found to be insensitive to the last parameter. Derivative-free optimization using a least squares method was found to be more efficient than the covariance matrix adaptation evolution strategy, which was also applied in earlier studies (Kriest et al., 2017; Kriest, 2017).

In general, these and other studies have found that local and gradient-based methods fare poorly in the optimization of BGC models. Ward et al. (2010) tested an adjoint-based gradient descent method against a micro-genetic algorithm, showing that both methods reduced misfits with observational data to similar extents, but the descent method could not consistently identify the same set of parameters; this outcome was attributed to under-determinism of the model. Athias et al. (2000) used TSEs to compare deterministic trust region, simulated annealing, and genetic algorithms, finding that the genetic algorithm was

the most reliable. Given the overall success of genetic and evolutionary algorithms, Mattern and Edwards (2017) compared four evolution-based optimization algorithms to calibrate two BGC models in a 3D modeling framework. All of the tested algorithms improved model results, even when truncating the optimization at roughly 100 model evaluations, but the artificial bee colony using differential evolution (a global, gradient-free method) performed the best.

Despite their successes, however, global and gradient-free methods can be prohibitively computationally expensive when estimating many uncertain parameters in BGC models coupled to physics-based representations of ocean mixing across multiple ocean locations. In some studies, the number of parameters estimated has been reduced to control the computational cost. For example, Kim et al. (2021) tracked 11 state variables in a BGC model with 72 parameters, but only 12 of the model parameters were ultimately estimated. This study also controlled cost by performing the estimation for a single site near the

West Antarctic Peninsula. Attempts have been made to perform parameter estimation across multiple sites, where the number of estimated parameters in each study was small (on the order of 10 parameters or less). Some studies used the optimized models to compare dynamics at contrasting sites (Ward et al., 2013; Kidston et al., 2011), while others tested the ability of BGC models to simultaneously represent different marine ecosystems (Hurtt and Armstrong, 1999; Friedrichs et al., 2007) or a larger region such as the North Atlantic (Schartau and Oschlies, 2003a).

Further attempts to overcome the computational cost of BGC parameter estimation include the use of physics-based surrogate models that represent realistic ocean mixing at a substantially reduced cost compared to 3D time-resolved simulations. For example, Kuhn and Fennel (2019) ran an ensemble of 1D models as a surrogate for a 3D simulation. This approach was used to calibrate and then compare three BGC models. The surrogate and full models produced similar errors with respect to observational data. The surrogate model recovered the average seasonal surface chlorophyll, and the BGC model implementa-

tion including temperature dependence performed better across the target region. By contrast, the most complex BGC model implementation was not able to represent variations in community structure across the region, making it a poor choice for application as a regional model. As an alternative to 1D surrogates, Kwon and Primeau (2006) used an off-line transport model developed from time-averaged velocity and diffusivity fields coupled to a BGC model to calculate phosphate equilibrium distributions. Using phosphate data with a gradient-free optimization algorithm, only one parameter could be constrained. In a

follow-up study, Kwon and Primeau (2008) examined carbon and alkalinity, but employed the same biophysical model. The transport matrix method has also been used to represent average advection processes in a computationally efficient manner (Kriest et al., 2017; Kriest, 2017; Sauerland et al., 2019).

  Ultimately, while previous attempts have been made to calibrate large BGC models, to simultaneously represent multiple sites, and to use physics-based surrogate models, the present study is the first where these three challenges are addressed

simultaneously. In particular, we outline a framework for estimating many uncertain parameters in a complex BGC model across a range of ocean conditions, using a physics-based model for vertical ocean mixing in a 1D water column configuration. We demonstrate the method for the 17-state-variable BGC Flux Model presented in Smith et al. (2021), which has 51 uncertain parameters. The parameter values provided in Smith et al. (2021) are taken as the baseline parameters in the present study, and the success of the proposed parameter estimation method is determined by the extent to which we are able to improve model

agreement with observational data, as compared to the baseline model. Using the 1D Princeton Ocean Model (POM1D) to

represent vertical mixing (Mellor and Yamada, 1982; Bianchi et al., 2006), we attempt to simultaneously match observational data for multiple state variables at two subtropical locations; namely, the sites of the Bermuda Atlantic time-series study (BATS; Steinberg et al., 2001) and the Hawaii Ocean time-series study (HOTS; Karl and Lukas, 1996). Because of the increased computational requirements imposed by the 1D mixing model, we focus primarily on computationally efficient gradient-based local optimization methods but still retain an initial global search to determine several regions of the parameter space in which to perform the local optimizations. The resulting framework is thus the first to combine both global and local methods in the context of a physically realistic, multi-site, multi-variable parameter estimation for BGC models with large numbers of uncertain parameters.

This paper provides a description of the proposed methodology for estimating model parameters in Section 2. Section 3 provides a description of the model and physical scenarios used to test the optimization routine. Section 4 includes the results of a TSE at the North Atlantic site and outlines the results of parameter estimations for the North Atlantic site, the Pacific site, and the two sites simultaneously. Finally, conclusions and a discussion of future work are included in Section 5. An appendix provides a discussion of the choice of state variables used during the optimization, the effects of different objective functions, and increasing levels of data sparsity.

## 2 Parameter Estimation Method

### 2.1 Problem definition

In the present study, we treat BGC model parameter estimation as an optimization problem, where we seek to minimize the error between observational data and the coupled BGC–vertical mixing model. We thus define a generic objective function, $\mathcal{J}$, corresponding to a particular choice of model parameters, $\mathbf{c}$, as the weighted sum of model error over the model state variables (e.g., phytoplankton, zooplankton, and nutrients) and ocean sites as

$$\mathcal{J}(\mathbf{c}) = \sum_{i=1}^{N_{\mathrm{s}}} \sum_{j=1}^{N_{\mathrm{v}}} \Pi_{ij} \delta_{ij}(\mathbf{c}), \tag{1}$$

where $N_{\mathrm{s}}$ is the total number of sites, $N_{\mathrm{v}}$ is the total number of state variables, $\Pi_{ij}$ is a scalar weighting factor, and $\delta_{ij}(\mathbf{c})$ is a function that describes the misfit between a model output and observational (or other reference) data for the $j$th state variable and $i$th ocean site. In the optimizations performed in this study, we use equal weights $\Pi_{ij}$ such that none of the target sites or variables are prioritized over others during the optimization. However, these weights can be adjusted to prioritize a subset of the state variables based on the specific aims of the user with no impact on the applicability of the method.

There are many possible ways to define the error function $\delta_{ij}(\mathbf{c})$, although herein we primarily use a normalized error based on the root mean squared difference (RMSD) between the model and observational data. In Appendix A, we consider the effect of different choices of $\delta_{ij}$, revealing little difference between the optimized results for the functions examined. The

RMSD-based error function is given by

$$\delta_{ij}(\mathbf{c}) = \frac{1}{\sigma_{ij}^{(\mathrm{obs})}} \left\{ \overline{\left[V_{ij}^{(\mathrm{obs})}(\mathbf{x},t) - V_{ij}(\mathbf{x},t;\mathbf{c})\right]^2} \right\}^{1/2} , \tag{2}$$

where $\sigma_{ij}^{(\mathrm{obs})}$ is the standard deviation of the observational state variable field $V_{ij}^{(\mathrm{obs})}(\mathbf{x},t)$ over all times and spatial locations (thereby capturing physical variability in the data, rather than observational uncertainties), $V_{ij}(\mathbf{x},t;\mathbf{c})$ is the corresponding modeled state variable field for parameter vector $\mathbf{c}$, and $\overline{(\cdot)}$ denotes an average over time and all available spatial dimensions.

The use of $\sigma_{ij}^{(\mathrm{obs})}$ in Eq. (2) is intended to normalize the RMSD between the model and observational fields, although different normalizations and formulations of the difference function are also possible, as is discussed at greater length in Appendix A. In the present study we demonstrate the parameter estimation method using time-resolved 1D (i.e., depth-dependent) observational and simulation data. However, the method can be readily extended to higher dimensional data sets and we thus leave the presentation as general as possible in the description of the approach.

Given the objective function in Eq. (1), we pose the parameter estimation problem as a constrained minimization of $\mathcal{J}(\mathbf{c})$, namely

$$\begin{aligned} &\min_{\mathbf{c}} \mathcal{J}(\mathbf{c}) \\ &\text{subject to } \mathbf{c}_{\min} \leq \mathbf{c} \leq \mathbf{c}_{\max}, \end{aligned} \tag{3}$$

where $\mathbf{c}_{\min}$ and $\mathbf{c}_{\max}$ are vectors of the minimum and maximum allowable values of $\mathbf{c}$, respectively. The final outcome of the estimation approach is a set of parameters, denoted $\mathbf{c}_{\mathrm{opt}}$, that minimizes the error over all state variables and ocean locations.

## 2.2   Optimization method

We use the open-source numerical analysis library DAKOTA (Adams et al., 2019) to solve the optimization problem defined in Eq. (3). The modular nature of DAKOTA provides a flexible framework for coupling an arbitrary model to a range of different numerical optimization algorithms, which the user can select and manage using control inputs. The modular structure of DAKOTA extends to the way in which it interacts with models, which are effectively treated as 'black boxes.' That is, the

user provides an interface that transfers the test parameters to the model, performs the simulation, and returns the objective function value to DAKOTA. Since the optimization algorithm does not have to be integrated into the model, interfacing the two is simple and non-invasive. The black-box approach is also ideal because DAKOTA contains various optimization strategies. Simply by editing an input file, we are able to use different optimization algorithms with little to no alteration of the model code.

Figure 1 provides a schematic of the coupling between DAKOTA and the biophysical model used to demonstrate the present parameter estimation approach (the model is described in Section 3). The dashed line in the schematic emphasizes that DAKOTA and the model do not interact directly; rather, the two are coupled through an interface script. An input file tells DAKOTA which numerical optimization tool to employ. DAKOTA produces a set of parameters to be tested and then runs the interface script. The interface script interprets the parameter values from DAKOTA and formats them appropriately for the model, which is then run. The simulation data is compared to the reference data to calculate and output the objective


function. DAKOTA reads the output and uses the data in its analysis routine, which continues to produce new parameter values until an optimal solution is found, according to user-specified convergence criteria. Parameter values are input to DAKOTA as normalized values $\tilde{\mathbf{c}} = (\mathbf{c} - \mathbf{c}_{\min})/(\mathbf{c}_{\max} - \mathbf{c}_{\min})$, such that $0 \leq \tilde{\mathbf{c}} \leq 1$. This normalization prevents the applied optimization algorithms from weighting parameters based on the relative magnitudes of parameter values. The normalized values are then re-scaled when the interface script interprets the DAKOTA output and sets up the model input files.

Leveraging the flexibility inherent in DAKOTA, we perform the parameter estimation using a hybrid optimization approach that incorporates both global (i.e., gradient-free) and local (i.e., gradient-based) methods. This hybrid approach is necessary to estimate a large number of uncertain parameters in complex BGC models while minimizing the required number of simulations, which can become expensive when the BGC model is coupled to a single- or multi-dimensional physical model and applied to various ocean locations. In total, there are three distinct steps in the present approach:

1. We randomly sample the parameter space $N_{\text{random}}$ times, run the biophysical model for each choice of parameters and evaluate the objective function $\mathcal{J}$ for each run. Because each of the model simulations is independent, this step can be easily parallelized and a large number of randomly selected parameter sets can be tested. The choice of $N_{\text{random}}$ will

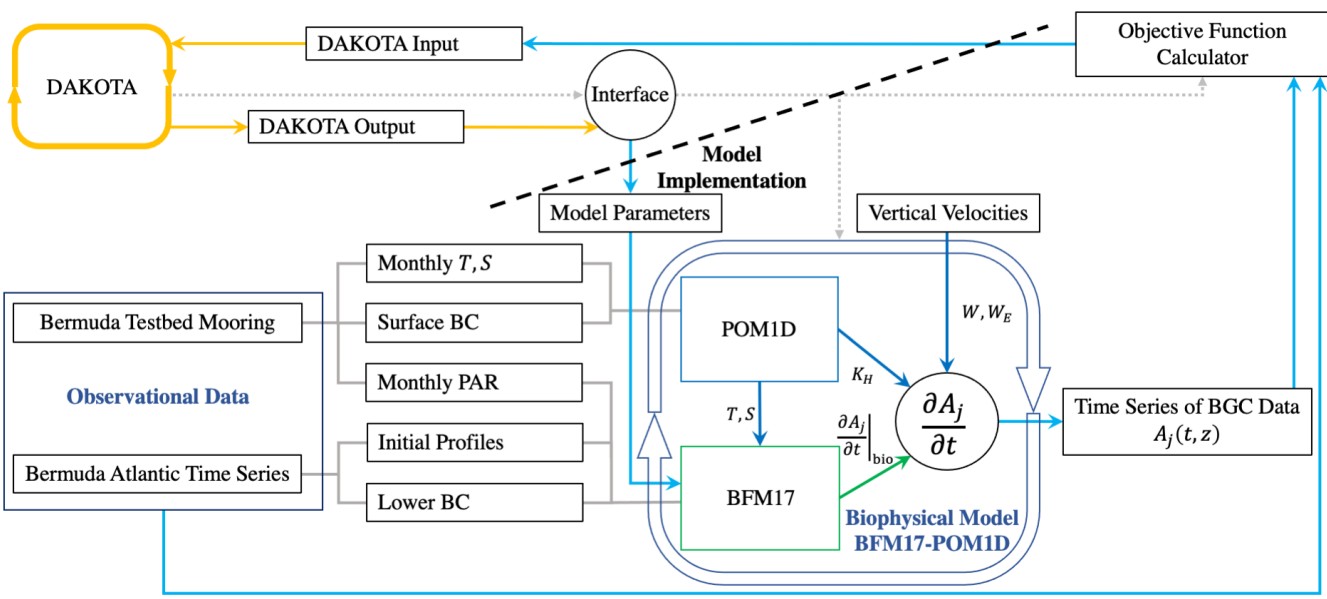

**Figure 1.** Schematic showing the coupling between DAKOTA, BFM17, and POM1D. The schematic shows observational data from BATS and the Bermuda testbed mooring, although data from HOTS are also used in the present study. The solid lines show the flow of information including parameter values and data. The grey dotted lines show how the different model components are run. DAKOTA calls the interface which evaluates the model then calculates the current value of the objective function. The coupled biophysical model, BFM17+POM1D, simulates the marine ecosystem for the corresponding site, producing a time series of vertical depth profiles for the 17 state variables, $A_j$. The target fields are compared to observational data with the objective value formatted so that it can be read directly by DAKOTA.

typically depend on the available computational resources, particularly for high-dimensional (in parameter space) BGC models.

2. We sort the $N_{\mathrm{random}}$ randomly sampled simulations based on the final values of $\mathcal{J}$ and use the parameter values from the $N_{\mathrm{top}}$ best cases to initialize a series of local gradient-based optimizations. The choice of $N_{\mathrm{top}}$ depends on the availability of computational resources as well as how quickly the values of $\mathcal{J}$ increase in the $N_{\mathrm{random}}$ sorted simulations.

3. We compare the final objective function values after gradient-based optimization for the $N_{\mathrm{top}}$ cases to determine a final set of parameters that gives the best agreement with observational data. The resulting best parameters are the final output of the parameter estimation method.

This hybrid multi-step approach combines the advantages of a global search with the computational speed-up enabled by a local gradient-based optimization. The initial global search does not guarantee that the method finds the global optimum. However, as we will show in Section 4, the method does provide good agreement between model results and multi-site observational data for a demonstration case with a 17-state-variable BGC model and two ocean locations. This provides confidence that the approach will also improve the agreement of other high-dimensional (in parameter space) BGC models.

To implement the first step of the method in DAKOTA, we used the Latin hypercube sampling algorithm to perform an efficient global search. For the gradient-based optimization in step two, a range of possible methods is available in DAKOTA. After testing various such methods, including the conjugate gradient method, we chose the quasi-Newton (QN) optimization algorithm included in the Opt++ library within DAKOTA. This is a C++ class library that uses object-oriented programming for nonlinear optimization (Meza et al., 2007). The QN algorithm reliably and efficiently converged to optimized solutions in our initial testing. In comparison, we found that the conjugate gradient method failed to converge efficiently, a result similar to that in the ecosystem parameter estimation study by Matear (1995). Based on results from test cases with smaller parameter sets (starting with five parameters), we attributed this slow convergence to excessively small steps required by the orthogonality constraint in shallow and elongated regions within the objective function space. Depending on the complexity and shape of the objective function space, it is possible that the conjugate gradient method may be feasible in other BGC model parameter estimations and the technique should not be discounted altogether.

Finally, we also explored the use of a genetic algorithm in the present parameter estimation method. However, for the 51-parameter BGC model that is the focus of the present demonstration, we estimated that a large and computationally expensive minimum population size of 366 members would be necessary. Because genetic algorithms cannot generally be parallelized to the same extent as Latin hypercube sampling, we were able to realize greater computational efficiency with the present approach. However, genetic algorithms do hold promise and could be incorporated with the present method in future work.

## 3 Biophysical Model and Physical Scenarios

### 3.1 Model description

We demonstrate the parameter estimation method described in the previous section with a BGC flux model that has 17 state variables and 51 free parameters, referred to as BFM17 (Smith et al., 2021). This model is a reduced implementation of the 56-state-variable BGC flux model, denoted BFM56 (Vichi et al., 1998, 2003, 2007). We couple BFM17 to POM1D to represent 1D depth-dependent physical mixing processes in the open ocean. A detailed description of the resulting coupled model, referred to as BFM17+POM1D herein, has been presented by Smith et al. (2021). Here we provide a summary of relevant model details

for the present work.

Figure 1 provides a schematic of the coupling between BFM17 and POM1D, where the total time rate of change of BGC state variables, $A_j$, at a given ocean location depends on both physical and biological processes according to

$$\frac{\partial A_j}{\partial t} = - \left[ W + W_E + v^{(\text{set})} \right] \frac{\partial A_j}{\partial z} + \frac{\partial}{\partial z} \left( K_H \frac{\partial A_j}{\partial z} \right) + \frac{\partial A_j}{\partial t} \bigg|_{\text{bio}} . \tag{4}$$

The first term on the right-hand side represents the vertical advection of tracers, which is parameterized by imposing a general

circulation vertical velocity profile, $W$, and an eddy velocity profile, $W_E$. The sinking of biological material is included as a constant advective velocity, $v^{(\text{set})}$. The second term on the right-hand side of Eq. (4) represents transport by small-scale turbulent diffusion, which is parameterized using the vertical diffusivity, $K_H$, calculated by POM1D. The final term is the time rate of change of $A_j$ from BGC processes, as determined from BFM17 using imposed temperature and salinity profiles corresponding to a particular site, as well as using data for the photosynthetically available radiation. Following the simulation

of BFM17 and POM1D, the coupler uses the velocities, vertical diffusivity, and the BGC time rate of change to determine the total rate of change of the 17 state variables in BFM17.

Both BFM17 and its larger precursor BFM56 use a chemical functional family (CFF) approach to model the marine ecosystem (Vichi et al., 1998, 2003, 2007). This approach provides a framework for easily controlling the complexity and specificity of the model by implementing different sets of phytoplankton, zooplankton, and nutrient groups. Each CFF is represented as

a vector of concentrations for the elemental constituents. For example, the phytoplankton CFF is a vector consisting of carbon, nitrogen, phosphorous, and chlorophyll concentrations. Each concentration corresponds to a state variable in the model. BFM17 was simplified to be a general and computationally cheaper model than BFM56 but retains the essential BGC processes for modeling a phytoplankton spring bloom (Smith et al., 2021). It is intended for future 3D simulations of upper, open-ocean dynamics at small scales, for example using large eddy simulations (Smith et al., 2016, 2018).

As described in more detail in Smith et al. (2021), BFM17 includes eight CFFs in total, comprising living organic, non-living organic, and inorganic groups. The two living functional groups (LFGs), phytoplankton, $P_i$, and zooplankton, $Z_i$, represent generic groups modeling the average community behavior of their respective populations. Dissolved organic matter, $R_i^{(1)}$, and particulate organic matter, $R_i^{(2)}$, are included as non-living organic CFFs. These four CFFs each have three components (indexed by subscript $i$), corresponding to the constituent elements carbon, nitrogen, and phosphorous ($i = $ C, N, or P, re-

spectively). The concentration of chlorophyll is included as an additional state variable for the phytoplankton LFG due to the

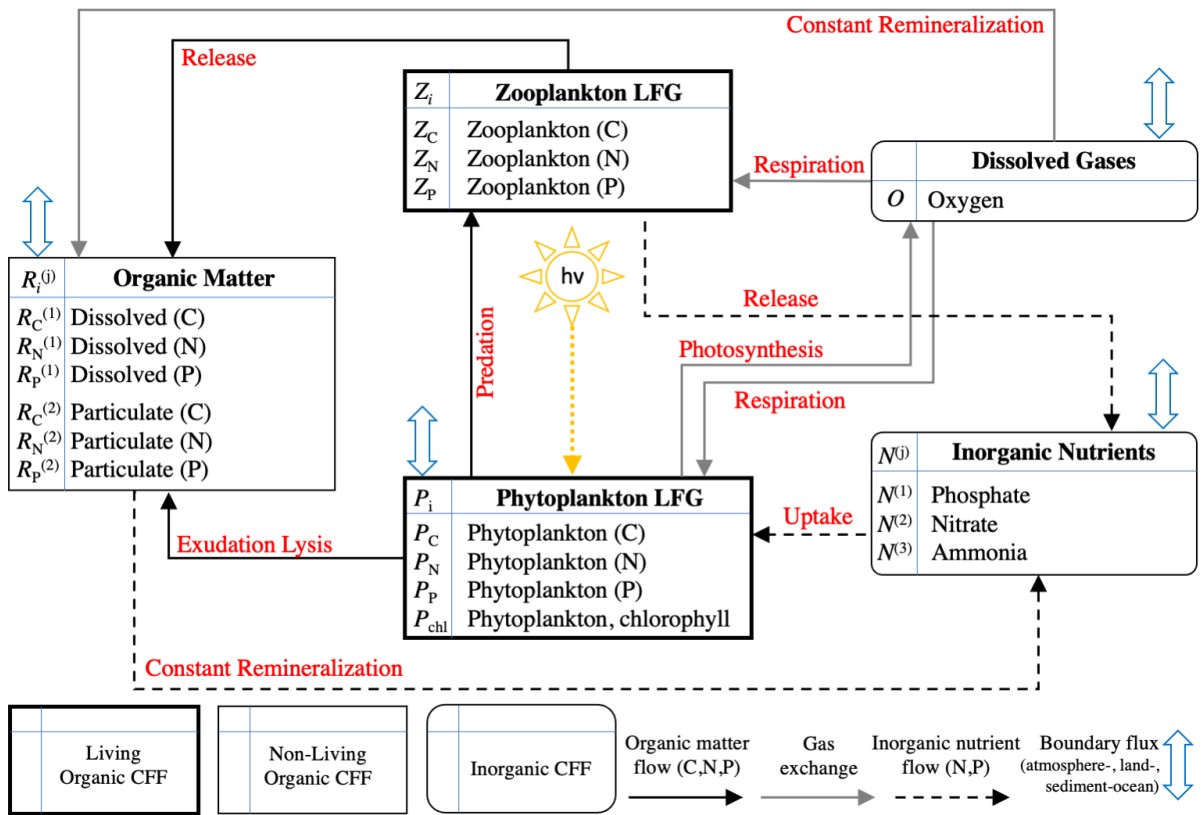

**Figure 2.** Schematic of the 17 variables included in BFM17 as well as the interactions between the variables (indicated by red text). The schematic is taken from Smith et al. (2021).

general interest in phytoplankton chlorophyll, the variable ratio to phytoplankton carbon concentration in the model, and the available observational data. The model includes three inorganic nutrients: phosphate, nitrate, and ammonia (denoted $N^{(1)}$, $N^{(2)}$, and $N^{(3)}$, respectively). Each nutrient only has a single state variable for its respective constituent component (P, N, and N for phosphate, nitrate, and ammonia respectively). The same is true of $O$, which is made up only of oxygen and is the only tracked dissolved gas.

Figure 2 shows the 17 state variables in BFM17 and illustrates fluxes of C, N, and P as parameterized in the model; Appendix C provides a more detailed description of the included processes. Table C1 in the appendix describes the 47 BGC parameters used in BFM17. Oxygen, phosphate, nitrate, and ammonia at the bottom of the domain are relaxed to observed concentrations; the four relaxation parameters, also included in Table C1, are included in the estimation of BFM17 parameters. The combined set of 51 parameters is thus the complete set to be estimated by the method outlined in Section 2. Smith et al. (2021) manually adjusted the 'baseline' parameter values in these tables to provide good agreement with data from BATS (Steinberg et al.,

2001). Our objective in the present study is to determine new parameter values that improve agreement with observational data at both the BATS and HOTS locations (Karl and Lukas, 1996).

The physical mixing model, POM1D (Blumberg and Mellor, 1987), uses vertical density profiles determined from observed temperature and salinity profiles to calculate the temporal evolution of horizontal velocities, the turbulent kinetic energy, and the mixing scale length, as described by Smith et al. (2021). The model then calculates the vertical turbulent diffusivity, $K_H$, included in the second term on the right-hand side of Eq. (4). The model uses the second-order turbulent closure model proposed by Mellor (2001), which is based on the model developed by Mellor and Yamada (1982) for the upper ocean.

## 3.2 Physical scenarios

We configured the coupled BFM17+POM1D biophysical model to simulate seasonal phytoplankton bloom dynamics measured at the BATS (Steinberg et al., 2001) and HOTS (Karl and Lukas, 1996) locations. Forcing and comparison data derive from monthly climatologies of BATS and HOTS observational data for the upper 150 m of the ocean to filter out interannual variability. At both sites, we assumed a 360-day climatological year with 12 months and 30 days per month. High-frequency forcing profiles were derived via interpolation of the monthly averaged observational data.

The observational state variables, $V_{ij}^{(\mathrm{obs})}(\mathbf{x},t)$, used in the error function, Eq. (2), have 150 discrete vertical values and 12 monthly averages. The corresponding model values, $V_{ij}(\mathbf{x},t)$, were obtained by monthly averaging the BFM17 state variables $A_j(\mathbf{x},t)$ at each of the ocean locations; the BATS data corresponds to $V_{1j}$ and the HOTS data corresponds to $V_{2j}$.

### 3.2.1 Bermuda Atlantic time-series study (BATS)

The mid-Atlantic implementation of BFM17+POM1D is based on observations from BATS and the Bermuda testbed mooring. Smith et al. (2021) discussed this implementation extensively. The observational data and model results for BATS are shown in Fig. 3. Here we compare five variables: phytoplankton chlorophyll, oxygen, nitrate, phosphate, and total particulate organic nitrogen (PON). The last of these variables, PON, is calculated as the sum of all nitrogen species from particulate organic sources as

$$\mathrm{PON} = P_\mathrm{N} + Z_\mathrm{N} + R_\mathrm{N}^{(2)}. \tag{5}$$

In our model, the organic sources of nitrogen include phytoplankton, zooplankton, and non-living particulate organic matter, corresponding to each of the terms on the right-hand side of Eq. (5). While Fig. 3 also includes model results after performing parameter estimations using the method described in Section 4, here we will only discuss the initial comparison between observational data (Fig. 3a) and the baseline implementation of BFM17 (Fig. 3b), where the baseline parameter values (summarized in Table C1) correspond to those from the manual estimation performed by Smith et al. (2021).

Observations collected at BATS reveal substantial seasonal variability in chlorophyll, while dissolved oxygen, nitrate, phosphate, and particulate organic nitrogen exhibit relatively uniform concentrations all year round (Fig. 3a). The seasonal climatology of chlorophyll measured at BATS shows a spring bloom that manifests most strongly in the upper ∼100 m of the water column in February, followed by elevated chlorophyll concentrations in the subsurface region (∼50-150 m) through the sum-

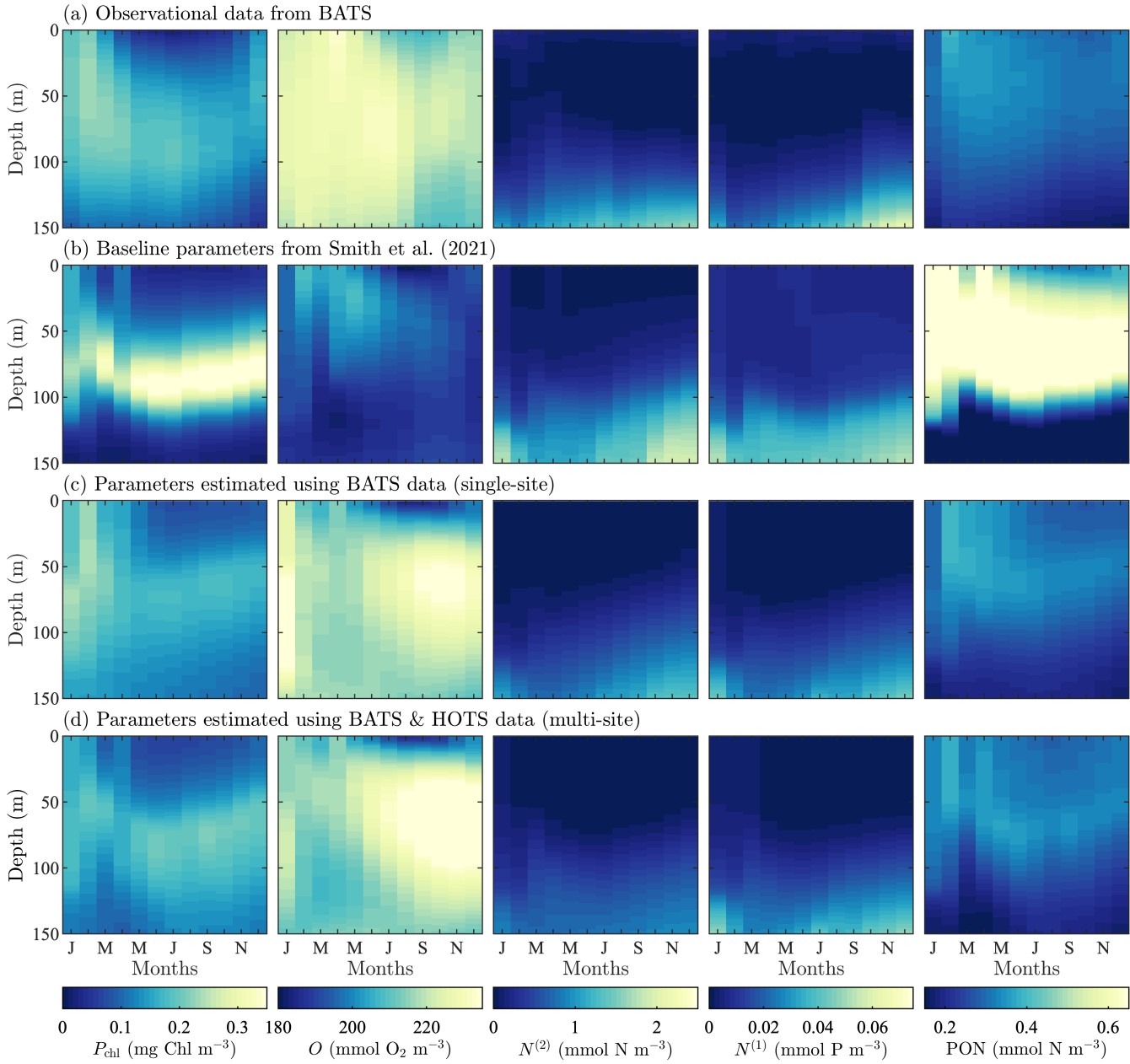

**Figure 3.** Phytoplankton chlorophyll ($P_{\mathrm{chl}}$), oxygen ($O$), nitrate ($N^{(2)}$), phosphate ($N^{(1)}$), and particulate organic nitrogen (PON) (columns from left to right) for the observational data from BATS (row a) and model results for: the baseline parameters defined in Smith et al. (2021) (row b), the parameters from the single-site parameter estimation (row c), and the parameters from the multi-site parameter estimation (row d). All data are shown as monthly-averaged depth profiles of state variable concentrations.

mer months (Fig. 3a). Oxygen has a near constant concentration from January through May, but during the fall a subsurface maximum develops between ∼25 m and 100 m. Nitrate and phosphate are confined to the bottom of the domain, while PON is confined to the upper portion of the water column with a slight deepening in the winter months.

Figure 3b shows that the baseline model phytoplankton chlorophyll results agree with the winter mixing and subsequent spring bloom in the observational data (Smith et al., 2021). The baseline model also captures the subsurface maximum of chlorophyll at roughly 100 m throughout the year. However, compared to the observational data, the baseline model over-predicts chlorophyll concentrations and has less vertical spread in the phytoplankton community through most of the year. The model also underestimates oxygen concentrations throughout the water column. The model data for nitrate and phosphate agrees with the observations fairly well, but with differences in the concentration magnitudes. Baseline model concentrations for PON are significantly higher than the observational data.

In summary, while the general bloom dynamics at BATS are captured by the baseline implementation of BFM17 from Smith et al. (2021), there is clearly room for improvement. Most notably, chlorophyll, oxygen, and PON model results could be improved, and these will be primary optimization targets for the parameter estimation method.

### 3.2.2 Hawaii ocean time-series study (HOTS)

A contrasting subtropical Pacific site was implemented in BFM17+POM1D using available observational data from HOTS. Figure 4a shows the observational data and Fig. 4b shows BFM17+POM1D results using the baseline model parameters from Smith et al. (2021). This figure also shows the fields from the single-site model calibration and the multi-site calibration, which will be discussed in Section 4. As with the BATS location, we focus on phytoplankton chlorophyll, oxygen, nitrate, phosphate, and PON. Since the baseline parameter values were manually tuned for the BATS location by Smith et al. (2021), there is no *a priori* expectation that these values will give good agreement between BFM17+POM1D and observational data at the HOTS location.

Observations collected at the HOTS location show fairly uniform chlorophyll, nitrate, phosphate, and PON concentrations across the seasonal cycle (Fig. 4a). Throughout the year, phytoplankton chlorophyll is elevated in the subsurface region in the upper ∼100 m. Oxygen has a maximum at the surface January through March, with a subsurface maximum developing May through November between ∼50 m and 100 m. Nitrate and phosphate are confined to the bottom of the domain with nutrient fluxes increasing from November through February. PON concentrations are highest in regions of the water column where phytoplankton and zooplankton thrive. By contrast to trends at BATS, PON concentrations increase from summer to fall at HOTS.

The baseline model predicts similar temporally uniform distributions of chlorophyll, nitrate, phosphate, and PON as the observations (Fig. 4b). However, the model substantially overestimates the magnitude of the subsurface chlorophyll maximum, and places it at too shallow a depth (Fig. 4b). Results for oxygen also substantially differ between the model and observations. In particular, the observations show that oxygen enters the domain through the surface during winter and is mixed to lower depths throughout the year. The model, by contrast, predicts high oxygen concentrations very close to the surface. The model severely over-predicts nitrate concentrations throughout the domain, while modeled phosphate concentrations are closer to

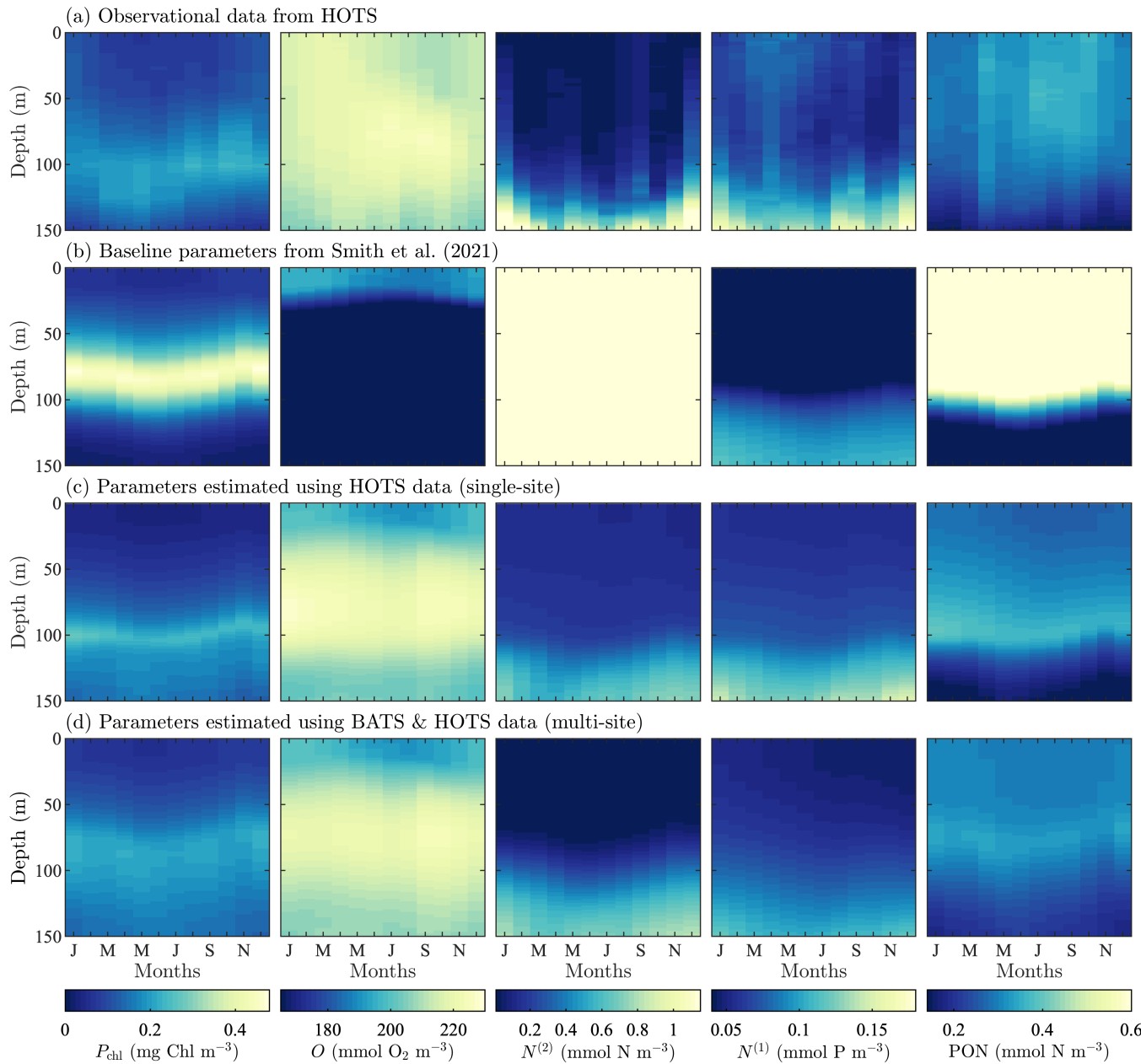

**Figure 4.** Phytoplankton chlorophyll ($P_{chl}$), oxygen ($O$), nitrate ($N^{(2)}$), phosphate ($N^{(1)}$), and particulate organic nitrogen (PON) (columns from left to right) for the observational data from HOTS (row a) and model results for: the baseline parameters defined in Smith et al. (2021) (row b), the parameters from the single-site parameter estimation (row c), and the parameters from the multi-site parameter estimation (row d). All data are shown as monthly-averaged depth profiles of state variable concentrations.

the observations (but still generally under-predicted). PON has slightly higher concentrations between 25 m and 75 m in the observational data, but the model over-predicts PON concentrations throughout the upper 100 m of the water column.

In summary, Fig. 4 shows that the baseline model generally agrees with the observational trends in phytoplankton chlorophyll, nitrate, phosphate, and PON, but the concentration magnitudes are over- or under-predicted to varying degrees. Both the trends and concentrations in the oxygen results differ substantially between the baseline model and observational data. Overall, these results indicate that the baseline BFM17 parameter values from Smith et al. (2021) are not well suited for the HOTS location. While this may be expected given that the baseline parameters were manually tuned to ensure reasonable agreement
with BATS observations (Smith et al., 2021), it motivates the search for new model parameters that simultaneously improve agreement with observations at the BATS and HOTS locations.

## 4   Parameter Estimation Results

### 4.1   Twin simulation experiment (TSE)

To verify the effectiveness of the parameter estimation method in reproducing known parameter values, we perform a TSE
using model-generated fields from BFM17+POM1D as reference 'observational' data. The reference model data are generated using baseline parameter values (Smith et al., 2021, summarized in Table C1). The objective of the TSE is to recover as many of the known parameters as possible using only state variable data from the model. For the reference run of the model, we simulated 30 days of data and stored daily values of all 17 state variables. We began the parameter estimation from initial parameter values perturbed upward by 10% from the baseline values, and we set the upper and lower parameter value bounds
(i.e., $\mathbf{c}_{\max}$ and $\mathbf{c}_{\min}$, respectively) to $\pm 25\%$ of the baseline values.

Figure 5 shows the combined results of the TSE and sensitivity analysis. The TSE results are shown as differences between the normalized parameter values at each optimization step, $\hat{p}_i$, and the normalized baseline value, $\hat{p}_o$. The parameters are normalized between 0 and 1 based on the upper and lower parameter bounds. Since the initial perturbation is calculated with respect to the standard (non-normalized) parameter value, the initial difference in the normalized parameter space is 0.2 for all
parameters. Of the 51 total parameters in BFM17, approximately 32 were recovered to within 5% of their respective baseline values, while 29 of those were within 1% of baseline values. Not all light and environmental parameters were fully recovered. However, the environmental parameters tend towards their baseline values, as do the phytoplankton parameters that were not fully recovered. Only some of the zooplankton parameters were successfully optimized, while the others did not change in value. All but one of the non-living organic parameters were successfully recovered.
To assess why certain parameters were not fully recovered, we performed a sensitivity analysis. For this analysis, we ran the coupled biophysical model with each parameter perturbed $\pm 5\%$ from the baseline value, excluding the parameters that exceed their respective standard bounds when perturbed. The standard bounds are those from Table C1, as opposed to the $\pm 25\%$ of the baseline values. Five parameters are only perturbed in one direction; $\delta_{Z,P}$ and $h_N^{(O)}$ were perturbed down 5% and $\zeta_{CO_2}$, $\zeta_{N^{(1)}}$, and $\zeta_{N^{(3)}}$ were perturbed up 5%. Parameters were compared using a sensitivity factor $S(p_i)$, defined as the
maximum objective function value from the evaluation of the positive and negative 5% perturbation cases for each parameter.

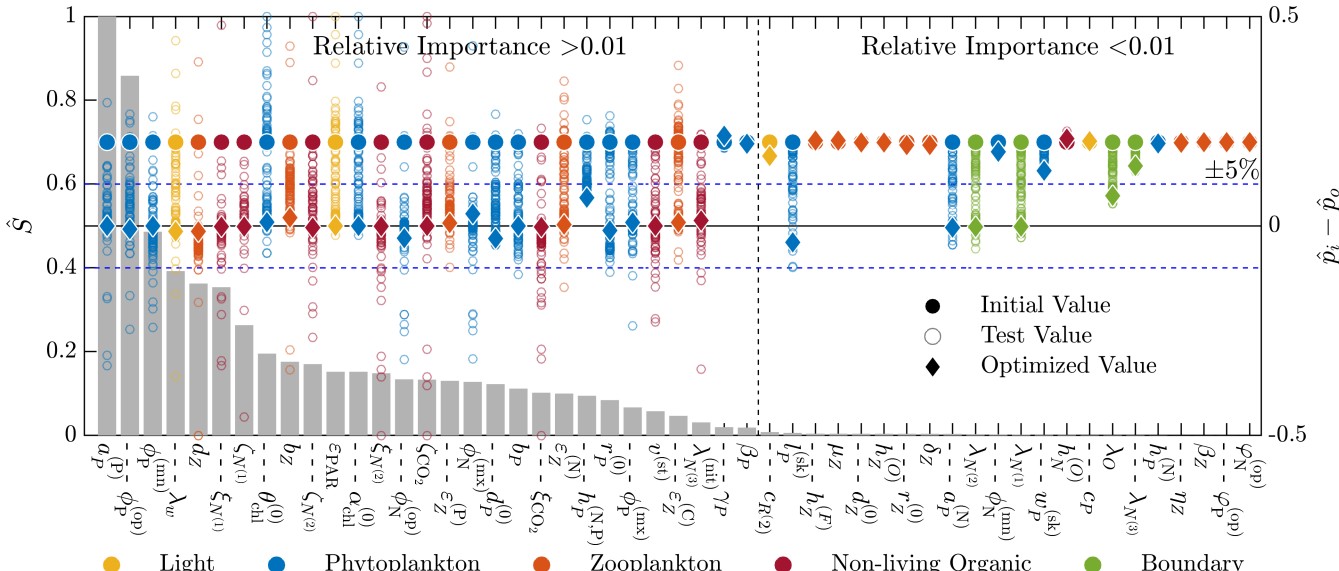

**Figure 5.** Results of the 51-parameter TSE and a single-perturbation sensitivity analysis. The TSE results show $\hat{p}_i - \hat{p}_o$, the difference between the test and baseline normalized parameter values, over the course of the optimization. Initial and final values are indicated by a circle and diamond respectively, and different colors indicate different parameter types (as noted in the legend at the bottom). The solid black horizontal line corresponds to the baseline parameter values and the dashed blue horizontal lines show $\pm5\%$ around the baseline. The relative importance $\hat{S}$ for each parameter, defined in Eq. (6), is shown using the bar graph in gray. This TSE is run with a truncated parameter space and only the first 30 days are used in the objective function.

The sensitivity factors were normalized by the maximum sensitivity factor

$$\hat{S}(p_i) = \frac{S(p_i)}{S_{\max}} , \tag{6}$$

where $S_{\max} = \max_i S(p_i)$. The values of $\hat{S}(p_i)$ provide an indication of the relative importance of the different parameters in changing the objective function. That is, lower values of $\hat{S}(p_i)$ indicate that changes in the parameter $p_i$ will have a small effect on the model error. As shown in Fig. 5, most of the parameters not recovered have a relative importance less than 0.01, as measured using $\hat{S}(p_i)$, or sensitivities less than 1% of the most sensitive parameter, which is the specific affinity constant for phosphorous $a_P^{(\mathrm{P})}$.

These results indicate that the parameter estimation method was successful in recovering the most sensitive parameters, while the least-sensitive parameters were not fully recovered. The optimizer and model correctly interface, and the optimization method performs as expected. With confidence in the optimizer and the interface, we performed an additional TSE that more closely mimics the calibration studies that will be performed for the BATS and HOTS locations. In this TSE, the synthetic reference data is different in two ways. First, the synthetic data is monthly averaged profiles of concentrations from the last year of daily data from a 3 year simulation using the baseline parameter values from Smith et al. (2021). Second, there are now only five quantities being used as target data: chlorophyll, oxygen, nitrate, phosphate, and PON. The upper and lower bounds

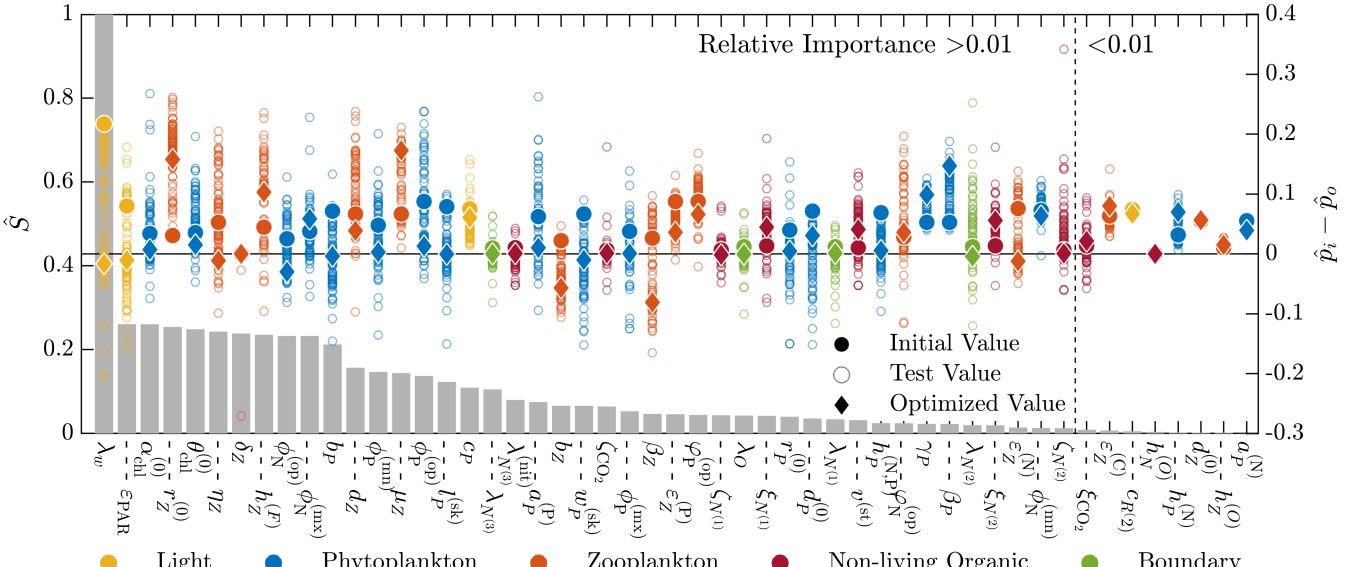

**Figure 6.** Results of the 51-parameter TSE and a single-perturbation sensitivity analysis. The TSE results show $\hat{p}_i - \hat{p}_o$, the difference between the test and baseline normalized parameter values, over the course of the optimization. The upper and lower bounds applied in this TSE correspond to those in Table C1. Initial and final values are indicated by a circle and diamond respectively, and different colors indicate different parameter types (as noted in the legend at the bottom). The solid black horizontal line corresponds to the baseline parameter values. The relative importance $\hat{S}$ for each parameter, defined in Eq. (6), is shown using the bar graph in gray. This TSE is run with the full parameter space and the objective function is calculated using the monthly average vertical profiles of five target fields from a three year simulation.

are set to the values included in Table C1 and we began the estimation from parameter values that are perturbed by $+10\%$ from the baseline. If the perturbed value exceeds the upper bound, the perturbed value is instead set to the upper bound.

Figure 6 shows the results of this additional TSE, where the parameter ranking and ordering now correspond to the five field objective function formulation over an entire year. Since the normalization is no longer a constant percentage of the nominal value, the initial shift in the normalized parameter space is not constant (as is observed in Fig. 5). We are able to recover the

baseline reference parameter values across the range of relative importances, but there are some parameters which fail to reach the baseline. Generally, those that do miss the target are trending towards the baseline value. Most final values are relatively close to the initial value, demonstrating the importance of finding appropriate initial parameters for the gradient-based local optimization step.

There are several other important differences between the results of this TSE and the prior TSE summarized in Fig. 5.

Overall, the number of parameters with relative importances greater than 0.01 increased from 29 to 43. Many more zooplankton parameters, in particular, become important, reflecting the importance of zooplankton in the annual cycle. The ranking of the parameters based on the relative importance also changed significantly, with the light parameters becoming the most important in the annual TSE, along with the aforementioned increased importance of the zooplankton parameters. Finally, the most

important parameter in the annual TSE, corresponding to the background attention coefficient of light, $\lambda_w$, is roughly four times more sensitive than any other parameter, a much larger difference than in the sensitivities for the 30 day TSE.

To further disentangle the reasons for the differences between the TSEs shown in Figs. 5 and 6, we performed two additional TSEs (not shown here) that used all 17 state variables in the objective function with the full parameter bounds from Table C1. In the first case we simulated 30 days only, similar to the TSE shown in Fig. 5 where we used tighter parameter bounds. In the second, we calculated monthly averages for the last year of three-year model runs, such that the resulting TSE can be compared to the results in Fig. 6 where only five state variables were used in the objective function. The resulting recovery of baseline parameter values for the two additional TSEs is qualitatively similar to those shown in Figs. 5 and 6 respectively. These results suggest that the duration of the model runs, more so than the parameter bounds or number of state variables, may be responsible for much of the difference between Figs. 5 and 6. With respect to the longer duration model runs, we also reiterate the possibility that the previously discussed issues of data sparsity and parameter dependence may contribute to a decrease in the number of fully recovered parameters.

Ultimately, since we are able to recover parameter values in both TSEs across the range of sensitivity values, we do not exclude any parameters in the subsequent model calibrations using data from BATS and HOTS. In general, the results from the TSEs provide confidence that the proposed optimization method will be able to drive the model parameter values in the direction of improved agreement with the observational reference data.

## 4.2 Single-site parameter estimations

For the single-site parameter estimations at both the BATS and HOTS locations, we used the method described in Section 2 to decrease the error in the $N_v = 5$ target fields corresponding to phytoplankton chlorophyll, oxygen, nitrate, phosphate, and total particulate nitrogen from organic sources. The method is applied using $N_{random} = 25,000$ random samples in the initial global search. We then began the local gradient-based optimizations from the $N_{top} = 20$ best samples. The value for $N_{random}$ was based on the availability of computational resources, while the value of $N_{top}$ was based on the rate at which the error increased from the best case. In particular, by the twentieth-best parameter set, the error increased by 19–30% among all randomly sampled cases at both locations. We thus assumed that $N_{top} = 20$ runs were sufficient as the increase in error was greater than $\sim$20%. The choices of $N_{random}$ and $N_{top}$ are problem- and resource-dependent and the present values should not be taken as fundamental to the method. Since we are treating the model calibration as a constrained optimization problem, parameters were only allowed to vary in the ranges included in Table C1. The computational cost of each parameter estimation (including the multi-site estimation in Section 4.3) is outlined in Appendix B.

Finally, it should be noted that some optimization and parameter estimation studies include replicate experiments. However, we have chosen not to do such experiments in the present study because of the random nature of the initial global search in the 51-dimension parameter space considered here. That is, with only 25,000 samples in the Latin hypercube step, it is highly likely that we will start the second-phase gradient-based optimizations in the replicate experiments from a completely unique set of parameter values, resulting in different final parameter values. Based on the following parameter estimation results, however,

**Table 1.** Normalized RMSD values ($\delta_{i,j}$) for the parameter estimation studies, calculated according to Eq. (2). The five target fields have normalized RMSD values for simulations performed with the baseline and calibrated parameter sets.

| Target field | BATS | | | HOTS | | |
|---|---|---|---|---|---|---|
| | Baseline case | Single site opt. | Multi-site opt. | Baseline case | Single site opt. | Multi-site opt. |
| Chlorophyll | 1.84 | 0.49 | 0.76 | 2.81 | 0.82 | 0.63 |
| Oxygen | 5.48 | 1.78 | 2.28 | 27.20 | 1.92 | 1.77 |
| Nitrate | 0.75 | 0.35 | 0.41 | 130.47 | 0.55 | 0.50 |
| Phosphate | 0.82 | 0.33 | 0.38 | 1.43 | 0.44 | 0.68 |
| PON | 5.80 | 0.37 | 0.58 | 7.54 | 0.93 | 0.58 |
| Total | 14.69 | 3.32 | 4.41 | 169.45 | 4.66 | 4.16 |

it will be seen that the 25,000 samples in the initial global search are sufficient to ensure that the overall method gives better agreement with the observational data than the baseline values from Smith et al. (2021).

### 4.2.1 Bermuda Atlantic time-series study (BATS)

Figure 3c shows that the final model fields after single-site parameter estimation at the BATS location are substantially closer to the observational data than the baseline model. This is supported quantitatively in Table 1, where the total RMSD between the model and observational data is nearly a fifth that of the baseline model. The RMSD also decreased for each target field individually, with the greatest improvement in PON, where the RMSD improved by an order of magnitude.

Improvements from the baseline to the calibrated model are reflected in Fig. 7, which shows fields of the normalized absolute differences between the model and observational data. The absolute difference is normalized by the standard deviation and provides a field-based representation of the different variables used to compute the objective function. Figure 7 shows that the calibrated model has a smaller error than the baseline model at nearly all depths and for all months.

Taken together, Figs. 3 and 7 show that the single-site calibrated model more accurately captures the magnitude of the subsurface maximum in chlorophyll, without changing the overall seasonality of phytoplankton chlorophyll. The calibrated model has a generally higher concentration of oxygen throughout the domain, improving agreement with the observations. There is also improved agreement for nitrate and phosphate due to a reduction in the predicted concentrations throughout the domain. The largest differences in nitrate, during fall and winter, are decreased in the calibrated model results, but there is a slight increase in disagreement at the bottom of the domain from April through September. The decrease in phosphate concentrations leads to smaller differences throughout the water column in the eight months of the year. For September through December, there is still improved agreement above ~125 m, but below that there is a slight increase in normalized differences. Concentrations of PON in the upper water column are substantially reduced compared to the baseline results, leading to significantly better agreement with the observational data.

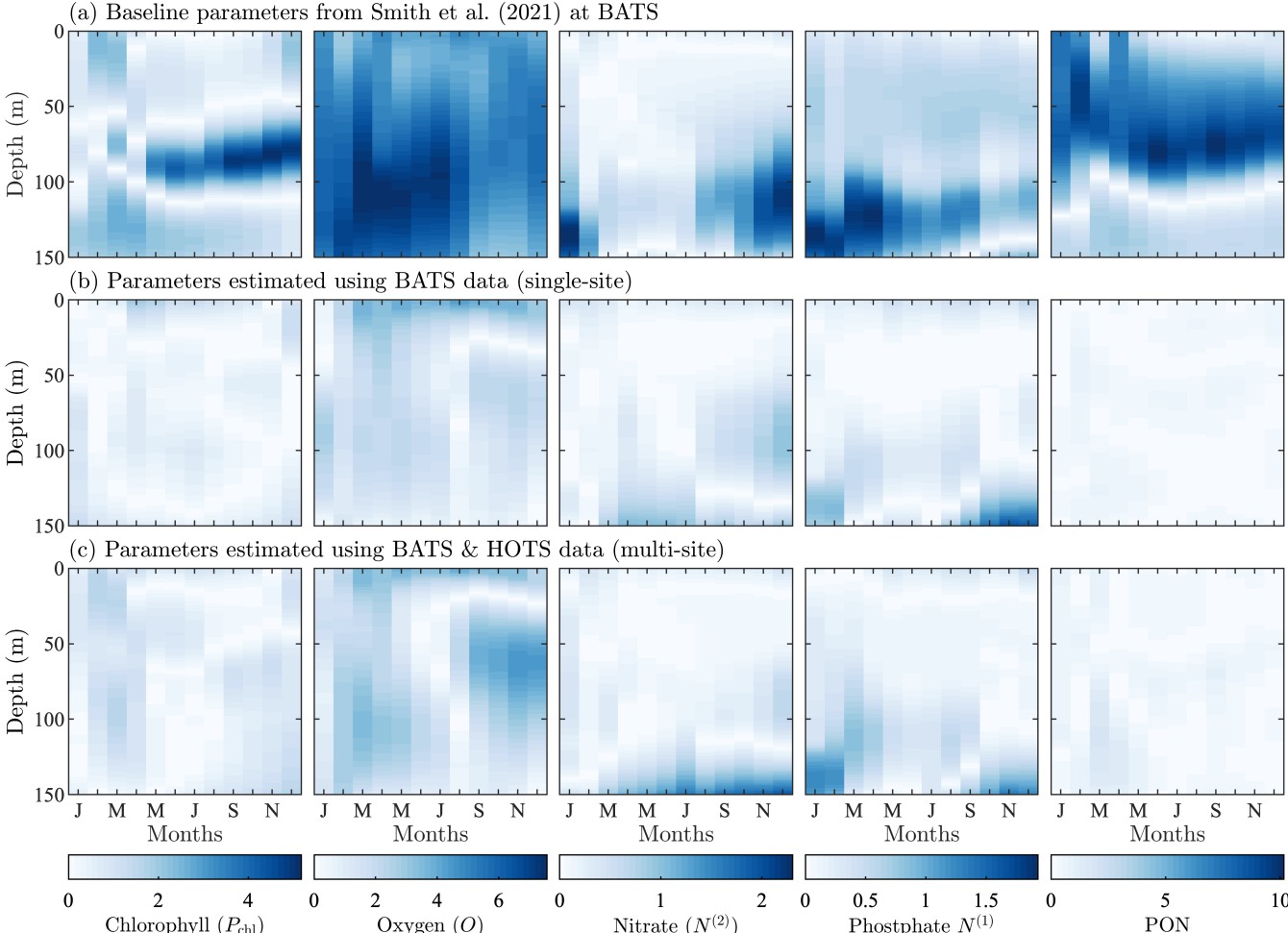

**Figure 7.** Normalized absolute differences between the monthly averaged field data from model runs and the corresponding observational data from the BATS site. The top row corresponds to the baseline parameter set from Smith et al. (2021), the middle row corresponds to parameters resulting from the single-site calibration, and the bottom row corresponds to parameters from the simultaneous calibration at the BATS and HOTS locations. The absolute difference values are normalized by the standard deviation of the corresponding observational field.

These results show that the parameter estimation method outlined in Section 2 was successful in improving the agreement between BFM17 and observational data at BATS. This is notable because the baseline model was itself manually tuned to give good agreement with the observational data (Smith et al., 2021), and the present automated method was able to produce even better agreement. Table C2 in the appendix provide the parameter values obtained for the single-site model calibration at the BATS location.

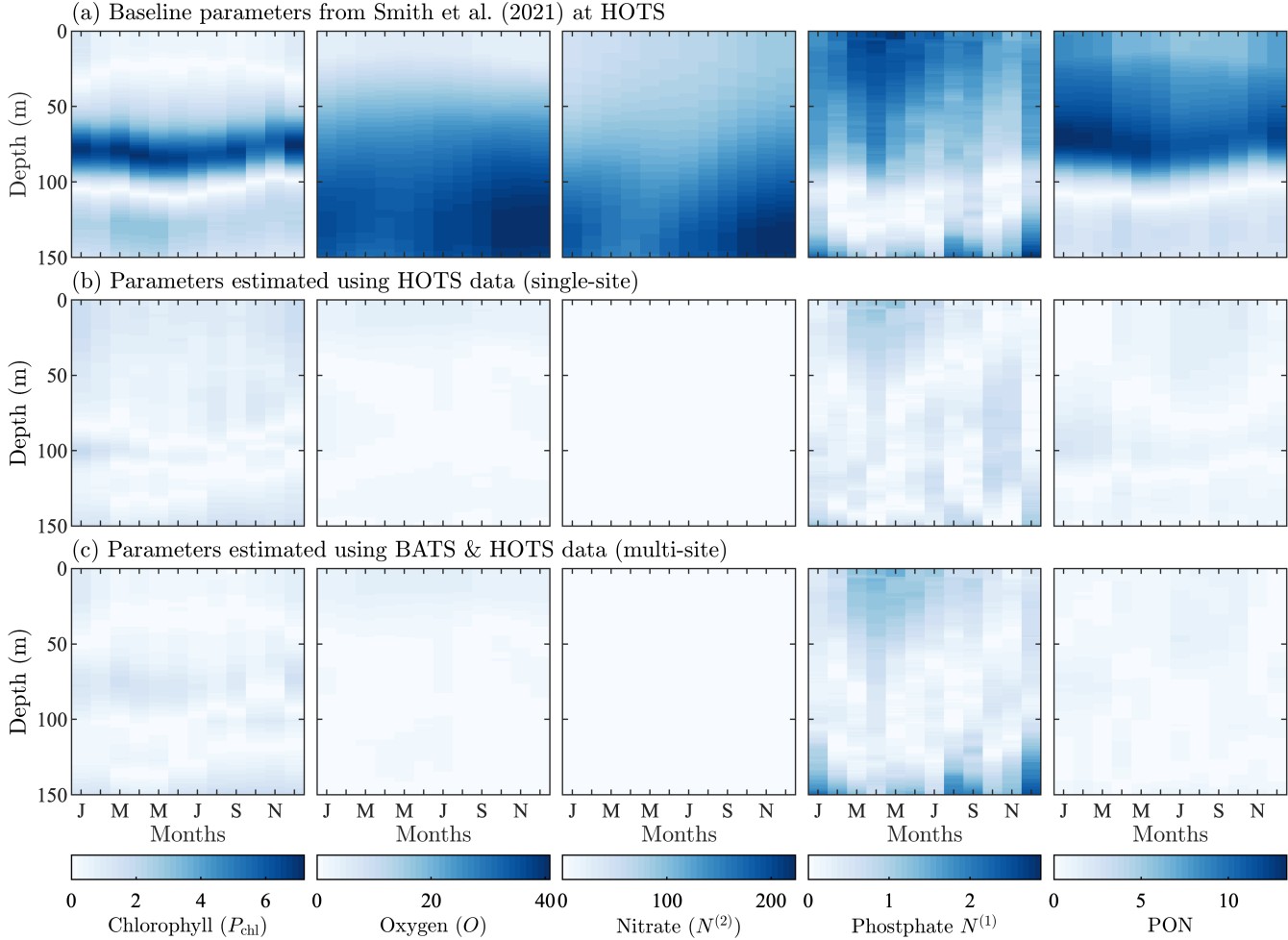

**Figure 8.** Normalized absolute differences between the monthly averaged field data from model runs and the corresponding observational data from the HOTS site. The top row corresponds to the baseline parameter set from Smith et al. (2021), the middle row corresponds to parameters resulting from the single-site calibration, and the bottom row corresponds to parameters from the simultaneous calibration at the BATS and HOTS locations. The absolute difference values are normalized by the standard deviation of the corresponding observational field.

### 4.2.2 Hawaii Ocean time-series study (HOTS)

For the parameter estimation at the HOTS location, Fig. 4 shows that the agreement between the model and observational data
increased substantially for all target fields, particularly compared to the very large errors from the baseline model (Smith et al., 2021) which has parameter values obtained from a manual estimation focused on reducing error at the BATS location only. Table 1 shows that the RMSD improved by a factor of between three (for phosphate) and over 236 (for nitrate). In total, the overall agreement with the observational data increased by a factor of nearly 36 from 169.45 for the baseline model to 4.66 for the calibrated model.

The difference fields in Fig. 8 further emphasize the improvement in agreement between the model and observational data at the HOTS location. For chlorophyll, there are large errors in the baseline model between 50 m and 100 m depths that nearly disappear in the calibrated model. The baseline model overestimates the subsurface maximum concentrations while underestimating the depth of the subsurface maximum. Both of these issues are resolved as the calibrated parameters result in lower chlorophyll concentrations and a deepening of the subsurface maximum. Although the calibrated model generally

produces better agreement for chlorophyll, the depth of the modeled subsurface maximum still shows more seasonality than is observed in the observational data. The subsurface maximum shoaled to above 100 m in the late summer and fall.

The normalized absolute differences for oxygen in the second column of Fig. 8 show notably improved agreement with observational data. As seen from a comparison of the fields in Fig. 4, the baseline model under-predicts oxygen, with the highest oxygen concentrations confined to the top of the domain. The calibrated model produces a more representative set of

oxygen concentrations, but the high oxygen concentrations remain between 50 m and 100 m all year round. The annual cycle in the observational data instead has high oxygen concentrations at the surface in the winter which then deepen such that there is a subsurface maximum near 75 m during the late summer and into the fall.

The calibrated model gives substantially improved agreement with nitrate observations, with the normalized RMSD decreasing from 130.47 for the baseline model to 0.55 for the calibrated model. The calibrated model also gives more accurate phos-

phate concentrations, with an annual cycle that includes more seasonality. The phosphate has increased bottom concentrations from November through January, similar to the higher concentrations for November through March seen in the observational data. Similar to all other fields, the calibrated model significantly improves predictions of PON, whereas the baseline model overestimates the observations by a factor of approximately three.

Overall, the parameter estimation method produced significantly better agreement with the observational data at the HOTS

location. Moreover, we were able to produce generally similar errors at the BATS and HOTS sites. Table C2 provides calibrated model parameters for the single-site estimation using data from HOTS.

## 4.3  Multi-site parameter estimation

We now use the parameter estimation method to calibrate parameters in BFM17 using observational data from the BATS and HOTS locations simultaneously. As with the single-site estimations, we performed an initial search of the parameter

space using $N_{\mathrm{random}} = 25,000$ samples and retained the $N_{\mathrm{top}} = 20$ best parameter sets to initialize subsequent quasi-Newton optimizations. The objective function for the multi-site parameter estimation was the summed normalized RMSD between model results and observational data for both sites. The optimization was performed without any weighting of the normalized error terms. That is, $\Pi_{ij}$ from Eq. (1) was unity for all $i$ and $j$. The fields with the most error therefore drive the optimization without any *a priori* determination of the relative importance of the fields or sites.

The resulting model fields for the BATS location from the multi-site calibration are shown in Fig. 3d, with the normalized differences between the model fields and observational data in Fig. 7c. Figures 4d and 8c show the resulting fields and differences for the HOTS location, respectively. Table 1 presents the field-specific and total normalized RMSD values for the calibrated model results.

Overall, the agreement between the multi-site calibrated model and the observational data at the BATS and HOTS locations is quite good, with errors comparable to results from the single-site estimations. The normalized combined model error was 184.14 for the baseline model, lowering to 8.57 after model calibration. At the BATS site, predictions for all of the target fields are closer to the observations for the multi-site calibrated model than the baseline model. For the HOTS site, the set of estimated parameters from the combined calibration improved agreement for all five target fields when compared to the baseline model and four of the fields when compared to the single-site optimization.

The multi-site calibration results have slightly larger errors than the single-site optimization at the BATS location. Figure 3 shows that the chlorophyll subsurface maxima are deeper in the water column than in the observational or single-site calibration data. The vertical skew also differs in the community structure where concentrations are dispersed further below the subsurface maxima, which is not observed in the observational data or the single-site calibrated model results. Oxygen results display a trend in the annual cycle similar to that observed in the single-site calibration, but concentrations in the subsurface maxima are larger during the later part of the year. Nitrate and phosphate concentrations are decreased further in the multi-site calibration, which produces larger differences at the bottom of the domain. Results for PON do not display the same build-up in particulate nitrogen in the beginning of the year, but generally have the same annual trend as the single-site calibration results.

Table 1 shows that the multi-site calibration results have similar error to the single-site optimization at the HOTS location, with the multi-site results actually displaying a slightly smaller total error than the single-site results. This counter-intuitive outcome is a by-product of the complexity of the 51-dimensional objective function parameter space, combined with the hybrid nature of the parameter estimation method proposed here. In particular, local gradients in the objective function space differ from the single-site case when including reference data from both BATS and HOTS, in this case permitting the gradient-based optimization to explore a broader region before reaching a convergence condition. Additionally, the initial random sampling is an important first step in the proposed hybrid approach but does not guarantee that a global minimum has been found in the objective function space. A larger number of initial random samples (i.e., larger $N_{\mathrm{sample}}$) would allow the method to probe new and potentially lower error regions of the objective function space, but we fixed this number at $N_{\mathrm{sample}} = 25,000$ in the present tests.

For the multi-site HOTS results, four of the five target fields (i.e., chlorophyll, oxygen, nitrate, and PON) are in better agreement with the observational data than the single-site results, with the only trade-off being increased error in phosphate. Chlorophyll, as observed in Fig. 4d, has more vertical spread with a less intense gradient around the subsurface maximum near $100\,\mathrm{m}$, which is also true of the observational data. Oxygen results for the multi-site case have higher concentrations throughout the domain, but not to the same extent as in the single-site results. The annual cycle in nitrate, phosphate, and PON are all similar to the observational data and the single-site results. In the nitrate and PON fields, lower predicted concentrations than those in the baseline and single-site calibrated models improve agreement with the observational reference data. The nitrate field, however, is underestimated at the bottom of the domain. Phosphate is the only field which, while still being better than the baseline case, has increased error compared to the single-site calibration.

Ultimately, although the present study serves primarily as a demonstration of the parameter estimation method, the parameter values from the multi-site calibration, summarized in Table C2, can be taken as the standard parameters for the BFM17 model,

replacing the baseline parameters outlined in Smith et al. (2021) when the model is used at new locations or in upper ocean
process studies.

While generally outside the scope of the present study, the optimized parameter values could be analyzed to better understand
the relationship between different ecological processes or the different sites. For example, the background attenuation coeffi-
cient, $\lambda_w$, which characterizes how murky the water is, has been minimized to 0.03 m$^{-1}$ from the initial value of 0.0435 m$^{-1}$ in
the three calibration cases. This suggests that the water is clearer than initially estimated. This result is reasonable considering
the initial value is based on parameter sets developed for coastal applications where the water would likely have more sedi-
ments blocking light. Another example is the relaxation diffusivity for ammonium at the bottom, $\kappa_{N(3)}$, which parameterizes
the return of ammonia from biological activity occurring at depth using the flux of PON. The calibrated diffusivity values are
all smaller than the initial value of 0.05 m$^2$ s$^{-1}$, which may suggest that there is a limited flux of ammonia. The nitrate flux may
therefore be sufficient to parameterize the nitrogen return from depth. Similar insights may be obtainable from the change in
other parameters during the optimization process, although this analysis quickly becomes complicated due to the large number
of parameters being calibrated. Nevertheless, the discussion of $\lambda_w$ and $\kappa_{N(3)}$ above demonstrates the way in which parameter
changes can yield new modeling insights.

## 5    Conclusions

We have formulated and demonstrated a method for simultaneously estimating a large number of uncertain parameters in com-
plex BGC models, considering multiple state variables and ocean locations. The method is fundamentally based on numerical
optimization whereby the error is reduced between model and observational (or other reference) data. Both gradient-free and
gradient-based optimization techniques are incorporated into the method to provide a broad exploration of the parameter space
combined with the computational cost savings enabled by local gradient-based approaches. While the broad search and mul-
tiple local optimizations do not guarantee that the solution is a global minimum, they do reduce the possibility of becoming
artificially trapped in regions of the parameter space based on inaccurate initial guesses by the user, while still taking advantage
of the computational efficiency of gradient-based methods.

As a demonstration of the method, we estimated the 51 parameters of BFM17 (Smith et al., 2021) using a 1D parameteri-
zation of open-ocean vertical diffusion from POM1D. We performed the estimation using observational data from BATS and
HOTS, both individually and together. In all cases, we were able to improve the model agreement with available observational
data, as compared to the manually tuned baseline model parameters for BFM17 provided in Smith et al. (2021). The result-
ing optimized parameters, summarized in Table C2, provide a more general implementation of BFM17 for use at new ocean
locations in the future. That is, the parameter set determined during the multi-site optimization should be treated as the most
globally applicable set of BFM17 parameters and should therefore be used in any future studies involving BFM17.

The present demonstration of the parameter estimation method is just one example of the many ways in which the method
can be configured. For example, given additional computational resources, a user may choose to expand the number of initial
random samples included in the gradient-free search of the parameter space or the number of subsequent gradient-based

local optimizations. Even for the relatively modest number of samples and local optimizations used here, we were able to significantly improve model accuracy. In Appendix A, we explore the impact of other choices in the method, including the use of only one state variable, alternative formulations of the objective function, and the omission of observational data, finding that the parameter estimation results were generally quite similar for these different choices. In future studies, the relative importance of the target fields or the relative confidence in the observed data can be used to weight the individual fields against each other.

Our proposed methodology also provides a general framework for sequentially probing parameter spaces in high-dimensional complex BGC models, followed by local optimizations. It can therefore be adapted in more substantial ways than simply changing particular optimizer configuration options. For example, while we found it computationally infeasible to run a genetic algorithm to convergence for this problem, truncated runs of that class of algorithms could be used instead of Latin hypercube sampling to identify multiple parameter sets that are then used to initiate local optimizations. This and other combinations of approaches are important directions for future study.

This study provides a method for determining the parameter values that provide the best possible fit to observational data, within the constraints of the dynamics represented by the BGC model itself. That is, the present method can be used to calibrate model parameters such that the dynamics represented in the model are the cause for any remaining data misfit. Previous studies have shown how model calibration can be used to determine the required set of dynamics (Hurtt and Armstrong, 1996, 1999; Friedrichs et al., 2007; Bagniewski et al., 2011; Ward et al., 2013). For example, Ward et al. (2013) removed fluxes between state variables by setting certain parameters to zero, thereby effectively determining not only the parameter values but also the model BGC pathways that could be excluded. In some studies, model calibrations that failed to sufficiently improve model results helped to identify deficiencies in the BGC models being employed (Spitz et al., 1998; Fennel et al., 2001; Schartau et al., 2001; Spitz et al., 2001; Schartau and Oschlies, 2003b). The models were either missing key BGC processes, such as key nutrient limitation or the bacterial loop, while others lacked fidelity in terms of the physical forcing. Application of the present method to studies of the dynamics included in BGC models is an important direction of future research.

Finally, the present approach can be extended to replace POM1D with a higher dimensional and more detailed physical model, such as a global circulation model (GCM). However, even with the cost savings enabled by a smaller BGC model such as BFM17, GCMs would still be extremely expensive to evaluate many tens of thousands of times, as is required even when using a gradient-based parameter estimation approach. It is common in optimization to use surrogate or lower-fidelity models to accelerate the optimization process, even when the intended application of the optimized parameters is a higher-fidelity simulation. In this sense, the current approach effectively employs POM1D as a physics-based, low-cost surrogate for a GCM.

## Appendix A: Alternative optimization configuration choices

There are a number of different ways that the parameter estimation method can be configured, with different choices of variables in the objective function, formulations of the objective function, and months included. In the following, we explore the effects of

each of these choices, with the understanding that the method outlined in Section 2 is intended to provide a general framework that is easily reconfigured as desired.

## A1 Parameter estimation based on chlorophyll only

Due to the specific interest in phytoplankton as a primary producer affecting both the carbon cycle and the food web, we tested single- and multi-site calibrations based exclusively on phytoplankton chlorophyll. Figure A1 shows the field results for this study, comparing chlorophyll fields from the observations, the baseline model, and single- and multi-site model calibrations based on five state variables (as in Section 4) and on chlorophyll only.

At both locations, Fig. A1 shows that the chlorophyll-only calibration recovers the observations to an even greater extent than calibrations based on five target fields. This improved agreement in chlorophyll is accompanied by reduced agreement in the other fields (not shown here). As with the multi-variable results, we again see that the single-variable results improve compared to the baseline model even when calibrating over two locations simultaneously.

The multi-site calibration results for chlorophyll show the way in which the parameter estimation method identifies parameters that balance the system behavior of the targeted communities. Comparing the single-site and multi-site calibration results, the predictions for the BATS location correspond to greater chlorophyll concentrations at depth, with suppression of phytoplankton growth at the beginning of the year. By contrast, chlorophyll at the HOTS location is concentrated higher in the domain with more seasonality and slightly higher concentrations. Ultimately, model results for one site are skewed towards the

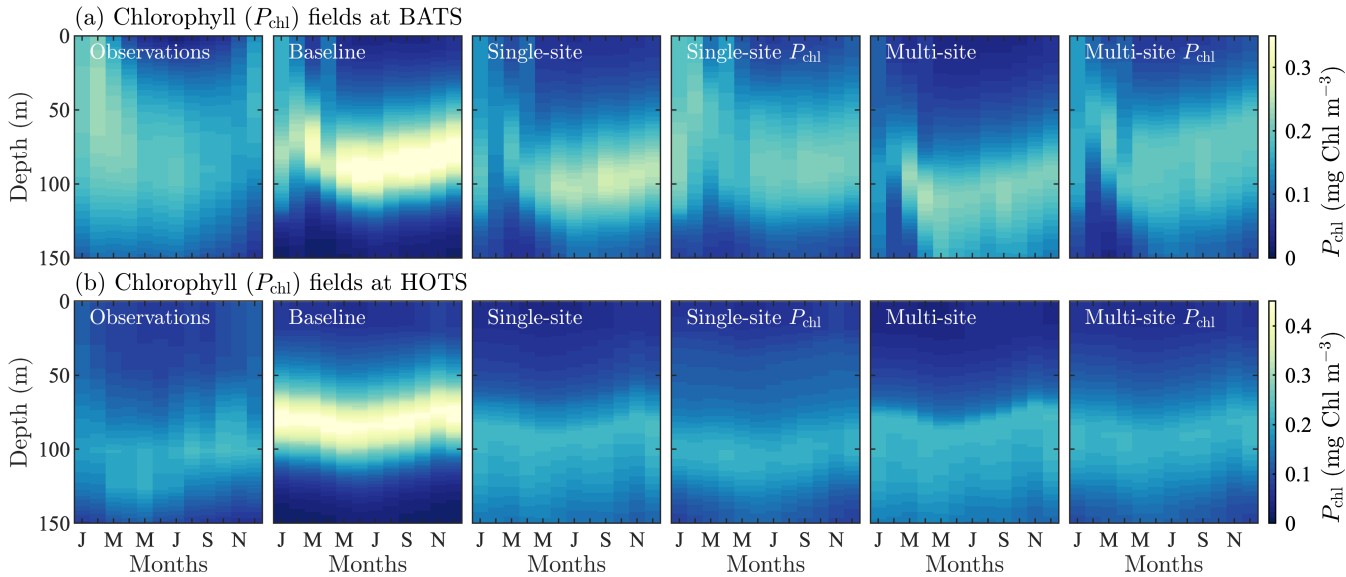

**Figure A1.** Phytoplankton chlorophyll ($P_{chl}$) for different optimization runs for the BATS and HOTS implementation of the model, top and bottom row respectively. The chlorophyll data is shown as monthly-averaged depth profiles. The fields show optimized results against observational data (first column) for the optimization case to compare having multiple objectives sites against optimizing only for chlorophyll.

behavior of the other site included in the calibration. Additional sites could be included in future work to obtain a more generic set of parameters.

## A2 Alternative objective function formulations

We next examine the impact of changing the objective function in the parameter estimation method, specifically by varying the original formulation of $\delta_{ij}$ from Eq. (2). Here we confine the analysis to single-site calibrations at the BATS location and test
three alternative formulations of $\delta_{ij}$.

The first alternative formulation multiplies the squared difference values by the reference values before being cube-rooted, namely

$$\delta_{ij}^{(A1)}(\mathbf{c}) = \frac{1}{\sigma_{ij}^{(\text{obs})}} \left\{ \overline{V_{ij}^{(\text{obs})}(\mathbf{x},t) \left[ V_{ij}^{(\text{obs})}(\mathbf{x},t) - V_{ij}(\mathbf{x},t;\mathbf{c}) \right]^2} \right\}^{1/3}. \tag{A1}$$

This particular form of $\delta_{ij}$ is intended to weight more where BGC processes are active in the water column, as represented
by higher concentrations of the state variables. Here, the weighting is applied by multiplying by the reference field so that the average incorporates the magnitude of the difference as well as the magnitude of the target concentration. In the second formulation, we modify the normalization factor on $\delta_{ij}$ to obtain

$$\delta_{ij}^{(A2)}(\mathbf{c}) = \left\{ \frac{\overline{\left[ V_{ij}^{(\text{obs})}(\mathbf{x},t) - V_{ij}(\mathbf{x},t;\mathbf{c}) \right]^2}}{\left[ V_{ij}^{(\text{obs})}(\mathbf{x},t) - \langle V_{ij}^{(\text{obs})}(\mathbf{x},t) \rangle \right]^2} \right\}^{1/2}, \tag{A2}$$

where $\langle \cdot \rangle$ denotes a time average. In this formulation, instead of normalizing with the standard deviation of all observational
data for a given field, the standard deviation is calculated at each spatial location (or depth, in the present 1D cases) relative to the time average only. This formulation accounts for the fact that the temporal variability in a given field can vary widely with spatial location and the overall standard deviation $\sigma_{ij}^{(\text{obs})}$ may not be a good representation of the variability at a particular location. Finally, the third formulation applies both modifications simultaneously. That is, the normalized error is calculated using

$$600 \quad \delta_{ij}^{(A3)}(\mathbf{c}) = \frac{\left\{ \overline{V_{ij}^{(\text{obs})}(\mathbf{x},t) \left[ V_{ij}^{(\text{obs})}(\mathbf{x},t) - V_{ij}(\mathbf{x},t;\mathbf{c}) \right]^2} \right\}^{1/3}}{\left\{ \left[ V_{ij}^{(\text{obs})}(\mathbf{x},t) - \langle V_{ij}^{(\text{obs})}(\mathbf{x},t) \rangle \right]^2 \right\}^{1/2}}. \tag{A3}$$

Model results after parameter estimation using the alternative formulations of $\delta_{ij}$ are shown in Fig. A2. Overall, the fields are all very similar, as well as similar to the results from the original formulation of $\delta_{ij}$ in Fig. 3. There are, however, differences in the predictions of phytoplankton chlorophyll and phosphate. Using the original formulation of $\delta_{ij}$, the phytoplankton chlorophyll is mixed throughout the water column from January through March. As spring continues, there is a more strat-
ified structure with a subsurface maximum that is vertically symmetrical and has decreasing concentrations away from the

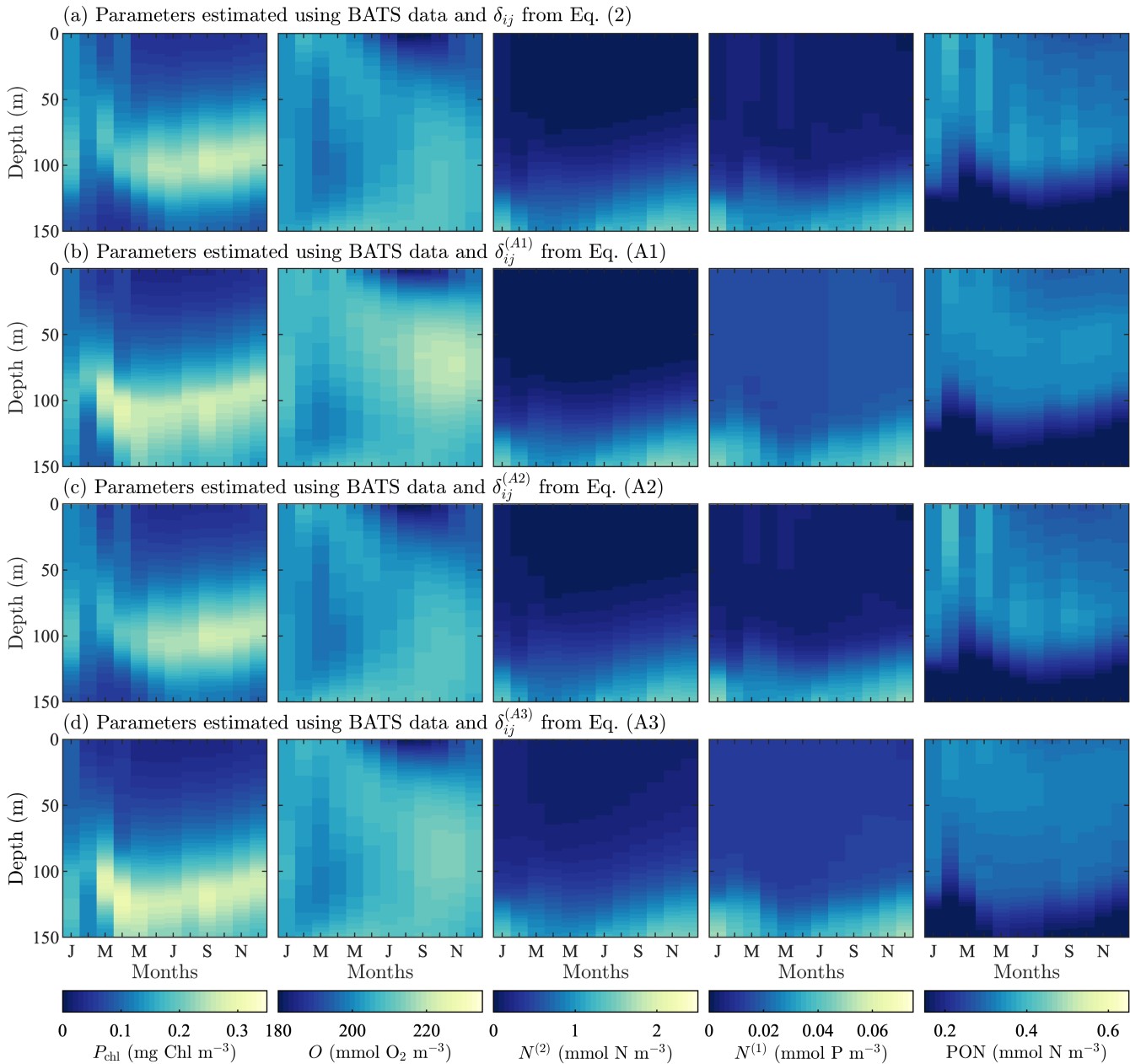

**Figure A2.** Phytoplankton chlorophyll ($P_{chl}$), oxygen ($O$), nitrate ($N^{(2)}$), phosphate ($N^{(1)}$), and particulate organic nitrogen (PON) (columns from left to right) for optimized results testing three alternative objective function formulations. The first (row a) tests using a cubed root formulation instead of the standard squared root difference from Eq. (1). Next, (row b) we test using a depth-averaged standard deviation instead of the standard deviation calculated using the entire time-depth field of the observational data. Finally, both changes to the objective function are tested simultaneously (row c). All data are shown as monthly-averaged depth profiles of state variable concentrations.

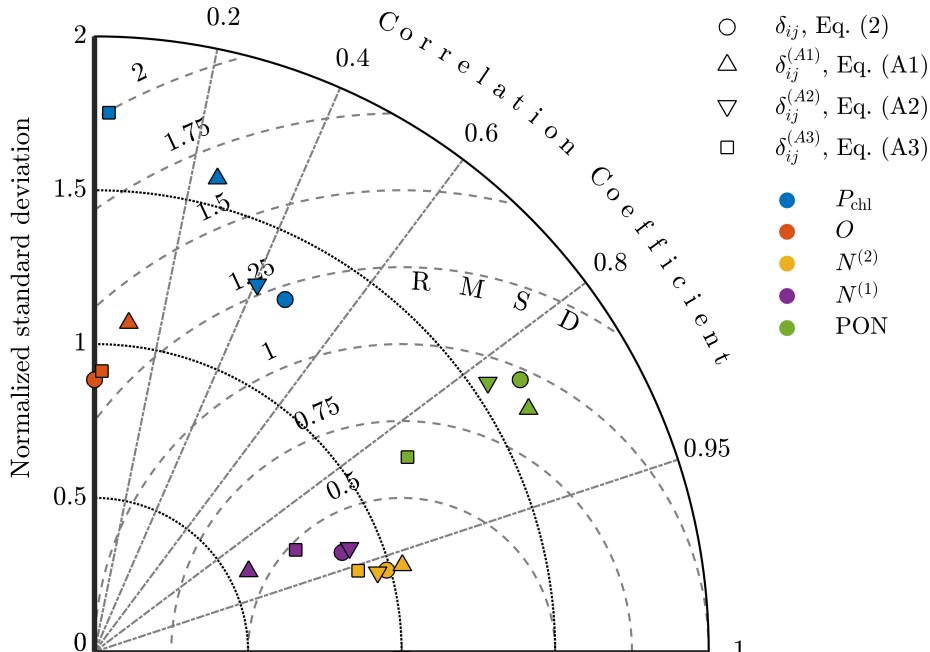

**Figure A3.** Taylor diagram comparing model results at BATS for phytoplankton chlorophyll ($P_{\mathrm{chl}}$), oxygen ($O$), nitrate ($N^{(2)}$), phosphate ($N^{(1)}$), and particulate organic nitrogen (PON) using parameter estimations with $\delta_{ij}$ from Eq. (2), $\delta_{ij}^{(A1)}$ from Eq. (A1), $\delta_{ij}^{(A2)}$ from Eq. (A2), and $\delta_{ij}^{(A3)}$ from Eq. (A3).

maximum. This is the same behavior produced by the second alternative formulation in Eq. (A2), but the other two formulations in Eqs. (A1) and (A3) broaden the region containing the subsurface maxima and break the vertical symmetry, with the phytoplankton decreasing less rapidly below the location of the subsurface maximum.

In the case of both oxygen and nitrate, the fields produced by each of the formulations of $\delta_{ij}$ are essentially the same. 610 There are small differences in the magnitude, but in terms of structure and concentrations, these differences are not significant. Phosphate, by contrast, is similar in that the structure of the calibrated fields is all the same, but the concentrations are less accurate when compared to observational data. The two alternative formulations in Eqs. (A1) and (A3) do not sufficiently decrease the field values leading to over-predicted concentrations in the upper portion of the domain.

The PON fields for all formulations of $\delta_{ij}$ have the same general structure, vertically and annually. The PON vertical structure 615 increases from an initial value to some maximum at depth, after which the concentration begins to decrease. The gradient is sharper below the maxima. The two alternative formulations in Eqs. (A1) and (A3) predict the maxima higher in the domain with a broader range of depths. The original formulation for $\delta_{ij}$ and the alternative formulation in Eq. (A2) predict the PON maxima to have a similar structure and depth as the chlorophyll subsurface maxima.

To compare the results quantitatively, Taylor diagrams with each of the alternative objective functions are shown in Fig. A3. 620 This diagram shows the normalized standard deviation, the normalized centered RMSD, and the correlation coefficient for

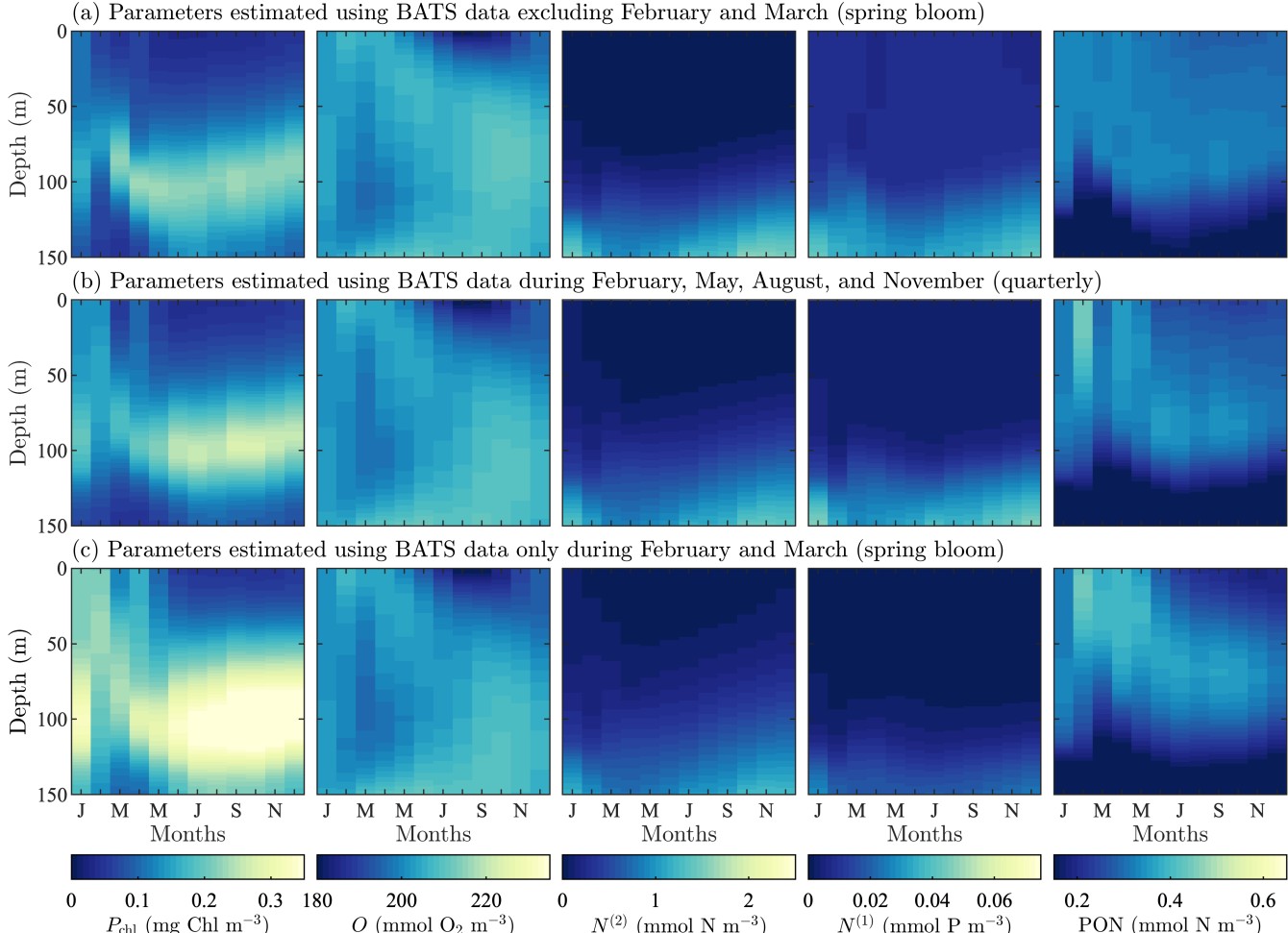

**Figure A4.** Phytoplankton chlorophyll ($P_{chl}$), oxygen ($O$), nitrate ($N^{(2)}$), phosphate ($N^{(1)}$), and particulate organic nitrogen (PON) (columns from left to right) for optimized results testing the effect of excluding data on the recovery of the annual trend in target fields. The data subset (row a) tests excluded the Spring Bloom data, February and March. Next, we test using quarterly observational data (row b). Finally, we test the extreme case of only having observational data from the Spring Bloom during February and March (row c). All data are shown as monthly-averaged depth profiles of state variable concentrations.

model results against observational fields. The point corresponding to the oxygen results from the calibration with Eq. (A2) is not shown here since it had a negative correlation coefficient of $-8 \times 10^{-3}$. The diagram demonstrates that the original formulation of $\delta_{ij}$ produces results for each of the fields that are either better or on par with the alternatives, providing confidence in this choice of $\delta_{ij}$ and indicating that the improvements in model accuracy outlined in Section 4 are robust to different

formulations of the objective function.

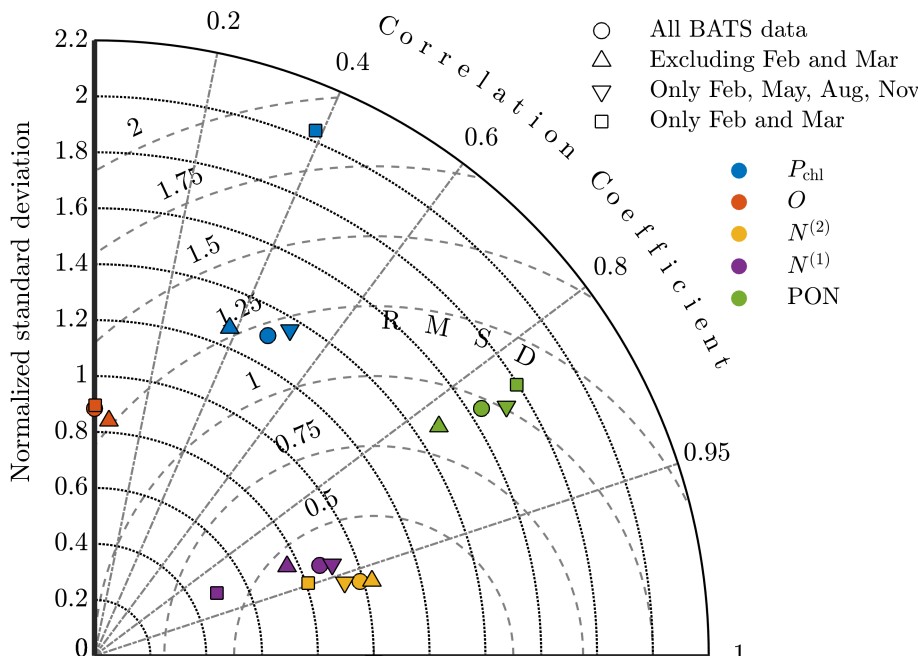

**Figure A5.** Taylor diagram comparing model results at BATS for phytoplankton chlorophyll ($P_{\text{chl}}$), oxygen ($O$), nitrate ($N^{(2)}$), phosphate ($N^{(1)}$), and particulate organic nitrogen (PON) using parameter estimations with all observational data, excluding February and March (i.e., all data except the spring bloom), including only February, May, August, and November (i.e., quarterly data), and including only February and March (i.e., only data from the spring bloom).

## A3 Data sparsity

To examine the effects of data frequency on the parameter estimation method, we performed three additional calibrations at the BATS location omitting data from two or more months during the parameter estimation (all three calibrations used five target fields in the objective function). In the first case, we examined the importance of capturing the initiation of the spring bloom by excluding all data for the months of February and March. This could be thought of as an experiment for data corruption considering the case where data from certain observational periods are unreliable and have to be excluded. In the next two cases we test realistic, if non-ideal, observation strategies where data is (*i*) collected quarterly in February, May, August, and November and (*ii*) only collected during the initialization of the Spring Bloom in February and March.

Figure A4 shows the resulting model fields after calibration. Oxygen, nitrate, phosphate, and PON fields all have consistent trends with only small differences, even when compared to the original calibration result from Fig. 3c. Chlorophyll does, however, demonstrate a higher sensitivity to data sparsity and the observational strategy employed. This is seen in the significantly higher concentrations in the phytoplankton chlorophyll subsurface maximum throughout the annual cycle. There are

also higher concentrations throughout the water column, particularly in phytoplankton growth from January through April. The increased activity corresponds to decreased temporal resolution.

The Taylor diagram in Fig. A5 further explores these results, comparing the standard single-site calibration study for BATS and each data sparsity case. Here we compare the models using the data set of the annual cycle for each of the target fields. The oxygen result for the quarterly observation strategy is not included since it has a negative correlation coefficient, but if plotted would be near the other oxygen results. The annual cycle for each of the five fields is represented to a similar level by the calibrated results for each case. The major exceptions are chlorophyll and phosphate in the case that only includes the spring bloom data. These fields have higher centered RMS error values and less representative standard deviations. It should be noted, however, that this calibration was based on data from only two of the twelve total months.

These results demonstrate that the annual cycle in the five target fields does not necessarily need to be observed on a monthly basis for optimization results to improve the model fit to the physical trends. Including the full data set did, however, produce the most representative parameter set. The results also highlight the danger of using data that are too sparse. In the data sparsity studies including and excluding the spring bloom, the spring bloom did not produce error measures consistent with the full data set. Calibrating using data only from the spring bloom led to good agreement between the included observational data and model results, but this came at the expense of not being generally representative of the annual cycle. These conclusions highlight the importance of matching the included data to the desired purpose of the optimized model and frequent—or at least even—coverage of the desired dynamics.

## Appendix B: Computational cost of model improvements

In this section, we briefly discuss the computational cost of running the parameter estimations presented in this study. All calibration studies are performed using the computational resources from the Cheyenne Supercomputer sponsored by the National Center for Atmospheric Research. The system features dual-socket nodes of 36 Intel Xeon processor cores. The BFM17+POM1D model is run using the system's GNU Fortran compilers, OpenMPI, and NetCDF.

The run time for a single model evaluation is approximately 5 min on a single core. DAKOTA provides the capability to perform multiple model evaluations in parallel. The total CPU time remains significant, but using supercomputing resources allows for drastic reductions in wall time. Table B1 includes details about the computer resources used and the CPU time for each calibration study. The data is split between the initial sampling and the local optimization steps of the optimization methodology and compared to the total CPU time. In the three parameter estimation cases above, the total CPU times are 31,052, 21,279, and 81,951 cpu-hours, with the initial sampling step accounting for 40%, 58%, and 55% of the total compute time for the BATS, HOTS, and multi-site calibration cases respectively.

To understand the relative improvement achieved by each stage of the multi-step calibration methodology, the results of the initial samplings and the optimization runs are included in Fig. B1. The first column shows plots of the twenty optimization runs compared to an optimization run from the baseline parameter values. The bar graph comparing the initial and final objective values demonstrate how much improvement we get for BATS, HOTS and the multi-site optimization in their respective rows

**Table B1.** The computational resources used to perform each parameter calibration study presented in Section 4. The sampling information is for the one sampling run used in the calibration methodology. The resource information for the optimization is reported on a per optimization run basis, while the reported CPU time is the total cpu-hours for running all 20 evaluations.

| Calibration Study | Sampling | | Optimization | | Full Methodology |
| --- | --- | --- | --- | --- | --- |
| | Resources | CPU Time | Resources per Opt | Accumulative CPU Time | CPU Time |
| BATS | 6 nodes 36 processor | 12,571 | 2 nodes 26 processor | 18,481 | 31,052 |
| HOTS | 6 nodes 36 processor | 12,247 | 2 nodes 26 processor | 9,032 | 21,279 |
| Multi-site | 10 nodes 36 processor | 45,000 | 2 nodes 26 processor | 36,951 | 81,951 |

in the figure. For reference, the second column of plots shows the evolution of the objective function for each optimization run. Note that the Y-axis has been cut off at 20, 30, and 50 and the X-axis at 300 evaluations for the BATS, HOTS, and multi-site optimizations runs respectively. Some values have been cut off, most notably the reference objective value, but these bounds most clearly show the convergence of the optimization runs. Objective values greater than the cut-off values are missteps which

the optimizer corrected. A few optimization runs continue beyond 300 model evaluations but none of them reach significantly different values from the one reached by evaluation 300.

For the BATS calibration case, the total normalized error is decreased by 31% comparing the best parameter set from the 25,000 sampled cases. By performing the subsequent local optimization runs we are able to get a 77% reduction in error. Simply doing a gradient-based optimization from the baseline parameter values gets us a 60% reduction in error, which is not

insignificant but is less than we were able to obtain.

For the HOTS site, we are able to get a 91% improvement in results due to the random sampling. This is in part simply due to how poorly we are initially performing at the HOTS site, since all previous work went into making the model representative of the BATS site specifically. This can also be seen in the number of sampled values which are better than baseline parameter set. For HOTS, 2,624 of the randomly sampled parameter sets are better than the baseline simulation, while for BATS only 188

random parameter sets are better than the baseline simulation. The optimization runs are able to further improve agreement by 71% from the best sampled case. A gradient-based optimization initialized from the baseline parameter set would reduce the accumulated normalized error to 8.94 which is higher than the 4.66 we are ultimately able to achieve by applying both pieces of the calibration methodology.

Similar to the HOTS case, the multi-site calibration is able to achieve significant improvement simply through the course

of the random sampling. The sampling produces 2,335 random parameter sets better than the baseline parameter set. The normalized error is decreased by 85% for the best performing randomly sampled case, as compared to the baseline case. This

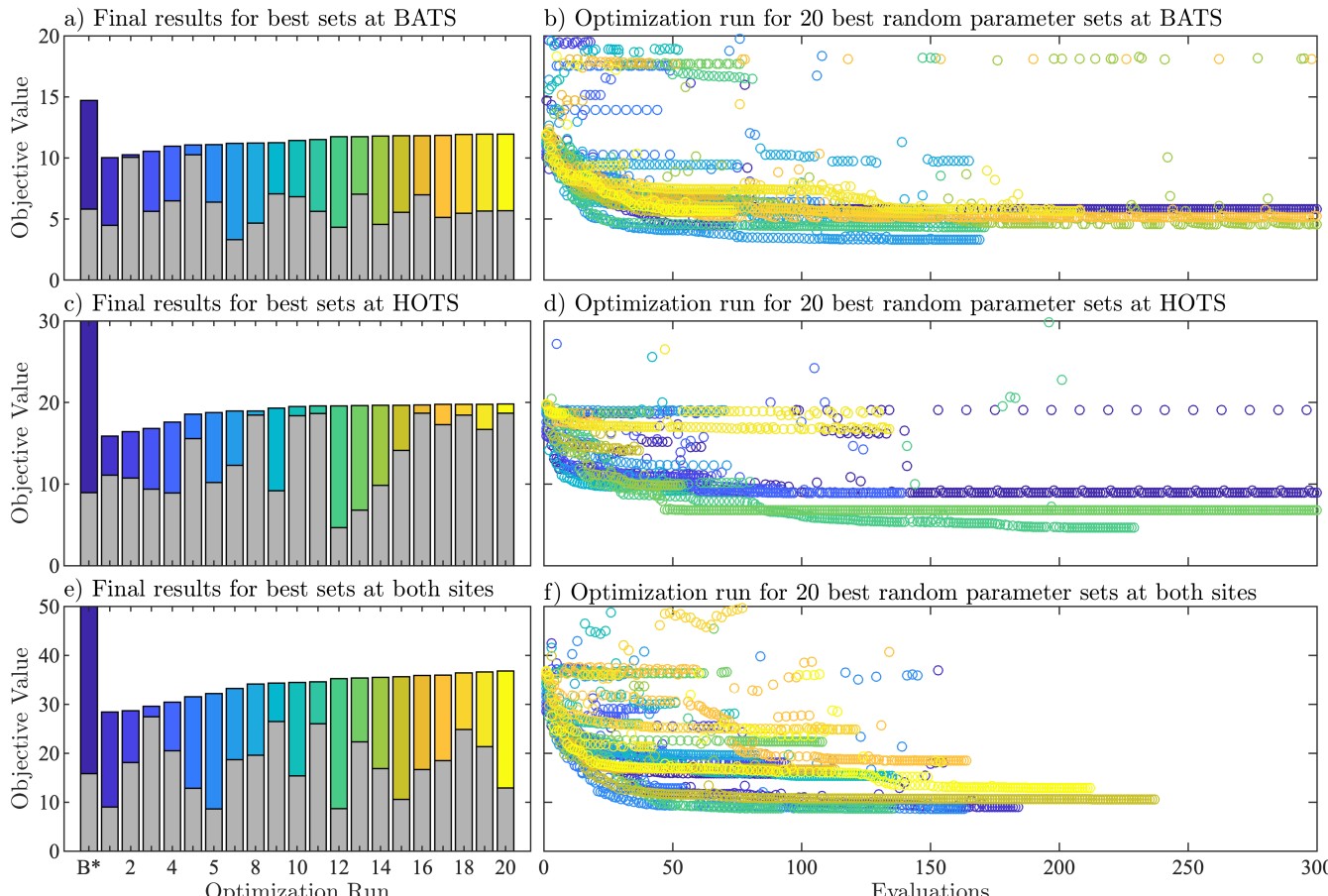

**Figure B1.** Results of the initial sampling and the optimization for each parameter estimation study: BATS (top row), HOTS (second row) and both sites (bottom row). The first column includes the initial (color coded) and final (gray) objective value for gradient-based optimization runs starting from the baseline parameters and 20 best sampled cases. The second column includes the objective function evaluation over the course of each optimization run. The max value of the baseline HOTS and multi-site calibration objective function have been cut off, being 169.45 and 184.15 respectively. Similarly, the axis of the optimization runs have been truncated to where most of the results can be seen. There are no significant improvements in the results for the cases run past 300 evaluations compared to the value reached at that point.

is still driven mainly by the excessive error in the HOTS field. By performing the optimization runs we are able to further decrease the error for an overall improvement of 95%, with the improvement from the best sampled case being 70%. The overall improvement of 95% exceeds the 91% which would be achieved just optimizing from the baseline model parameters.

In all cases, the use of the 20 best cases from randomly sampled parameter sets to initialize gradient-based optimizations resulted in improved agreement over that achieved by simply optimizing from the baseline cases. The proposed methodology has been developed in response to constraints of using local, but efficient, optimization methods. We have been concerned about the possibility of falling into local minimums in arbitrary regions of the parameter space. The baseline optimization

results having comparable objective functions values, even outperforming particular optimization from the sampled parameter sets, emphasizes that we cannot completely rule out the possibility that a parameter set with a higher objective value but in other regions of the parameter space could produce better agreement with observational data than we achieved with the 20 best sampled cases when optimized. In this work we assumed that selecting the sampled cases with the lowest error is the best way for identifying regions with relatively low error. Considering these results, it is worth determining in the future how this methodology could be modified to use a different criterion for selecting the parameter sets used to initialize the optimization runs. For example, instead of only using the error-based objective function, one could incorporate some measure of distance to the selection criterion to ensure that we are getting reasonable coverage of the parameter space. This is not done in this work but it is worth highlighting and should be considered for future work. The major challenge will be determining how to ensure sufficient coverage of the parameter space for such a high-dimensional parameter space. This challenge motivated the random sampling proposed in the current methodology.

## Appendix C:  Description of BFM17

Smith et al. (2021) provide a detailed description of BFM17, but here we outline the primary processes represented in the model. Phytoplankton gross primary production results from the consumption of carbon dioxide, nitrogen, ammonia, and phosphate during photosynthesis. Since carbon dioxide is treated as an infinite source, phytoplankton growth can only be limited by either the availability of nitrogen, phosphorous, or light. The model parameters controlling the availability of light are the first four parameters in Table C1. Phytoplankton losses include respiration, exudation, lysis (cell rupture), and predation by zooplankton. The metabolic activity of phytoplankton results in carbon losses during respiration. Following the breakdown of sugar, carbon dioxide is released from the cell. Carbon is also lost via exudation when there is not enough nitrogen or phosphorous for the carbon to be assimilated. In this case, the carbon is sent directly to the dissolved organic matter carbon pool. Phytoplankton matter is also lost to lysis, which can result from virus penetration of the cell membrane. Lysis fluxes matter to both the dissolved and particulate organic matter CFFs in terms of all three constituent components. Ultimately, the change in phytoplankton chlorophyll is calculated from the uptake of carbon by phytoplankton minus the losses from the previously described processes. The quantity is regulated by the ratio of realized to potential rates of photosynthesis.

The zooplankton LFG is treated as carnivorous and, consequently, the sole source of growth for the LFG is its predation on phytoplankton. As carbon, nitrogen, and phosphorous are lost by phytoplankton from predation, all three constituent pools increase for zooplankton. Zooplankton is a living organism so there are carbon losses resulting from respiration as part of the organism's metabolic activity. The zooplankton losses resulting from egestion, excretion, and mortality are parameterized as releases to the dissolved and particulate organic matter pools for all three constituent components. Nitrogen is also released to ammonia while phosphorous is released to phosphate.

As noted, the non-living dissolved organic matter increases from phytoplankton losses due to lysis and releases from zooplankton. The dissolved organic carbon also increases from phytoplankton exudation. Dissolved organic nitrogen can be lost as a result of phytoplankton uptake of nitrate and ammonium. Similarly, dissolved phosphorous can be lost as a result of phyto-

plankton uptake of phosphate. Non-living particulate organic matter has a more uniform behavior across the three constituent components. In all cases, the particulate matter results from the lysis by phytoplankton and the release of organic matter from zooplankton.

Instead of nonliving organic matter being recycled back to the inorganic nutrient pools through a bacterial loop, BFM17 uses a constant remineralization rate closure. Matter is cycled directly back to the inorganic nutrient pools based on the product of a constant rate and the non-living organic matter concentrations. Carbon is also remineralized back to carbon dioxide, but since the inorganic dissolved gas acts as a sink it is not being tracked in this model implementation.

Oxygen is the only dissolved gas that BFM17 explicitly tracks. Oxygen is introduced into the system via aeration of the

surface water resulting from wind forcing calculated with observational data. The production of oxygen by phytoplankton during photosynthesis is the only biological source of oxygen. Oxygen is consumed during phytoplankton and zooplankton respiration as well as the recycling of non-living dissolved and particulate organic carbon to carbon dioxide. Oxygen is also lost to nitrification, a process that converts ammonium to nitrate.

Phosphate, nitrate, and ammonia are consumed by phytoplankton. Phosphate and ammonia are replenished through the

release of phosphorous and nitrogen, respectively, by zooplankton. Phosphate and ammonia also receive matter from the remineralization of dissolved and particulate organic matter. Remineralization only returns nitrogen to the ammonium pool from which the nitrate pool is replenished via nitrification. During nitrification, nitrogen from ammonia is combined with oxygen.

**Table C1.** List of BFM17 parameters controlling the marine ecosystem dynamics in the model.

| No. | Parameter | Baseline value | Units | Range | Description |
|---|---|---|---|---|---|
| | | | **Phytoplankton Parameters** | | |
| 1 | $\varepsilon_{\mathrm{PAR}}$ | 0.4 | - | 0.25–0.75 | Fraction of photosynthetically available radiation |
| 2 | $\lambda_w$ | 0.0435 | $\mathrm{m}^{-1}$ | 0.03–0.05 | Background attenuation coefficient |
| 3 | $c_P$ | 0.03 | $\mathrm{m}^2(\mathrm{mg\,Chl})^{-1}$ | 0.005–0.045 | Chlorophyll-specific light absorption coefficient |
| 4 | $c_{R(2)}$ | $0.1\times10^{-3}$ | $\mathrm{m}^2(\mathrm{mg\,C})^{-1}$ | $1.5\times10^{-5}$–$1.5\times10^{-4}$ | C-specific attenuation coefficient of particulate detritus |
| 5 | $r_P^{(0)}$ | 1.6 | $\mathrm{d}^{-1}$ | 1.0–5.0 | Maximum specific photosynthetic rate |
| 6 | $b_P$ | 0.05 | $\mathrm{d}^{-1}$ | 0.005–0.075 | Basal specific nutrient-stress lysis rate |
| 7 | $d_P^{(0)}$ | 0.05 | $\mathrm{d}^{-1}$ | 0.005–0.075 | Maximum specific nutirent-stess lysis rate |
| 8 | $h_P^{(\mathrm{N,P})}$ | 0.1 | - | 0.005–0.15 | Nutrient-stress threshold |
| 9 | $\beta_P$ | 0.05 | - | 0.005–0.1 | Excreted fraction of primary production |
| 10 | $\gamma_P$ | 0.05 | - | 0.005–0.1 | Activity respiration fraction |
| 11 | $a_P^{(\mathrm{N})}$ | 0.025 | $\mathrm{m}^3\,(\mathrm{mg\,C})^{-1}\mathrm{d}^{-1}$ | 0.005–0.05 | Specific affinity constant for nitrogen |
| 12 | $h_P^{(\mathrm{N})}$ | 1.5 | $\mathrm{mmol\,NH_4\,m}^{-3}$ | 0.25–5.0 | Half-saturation constant for ammonium uptake |
| 13 | $\phi_N^{(\mathrm{min})}$ | $6.87\times10^{-3}$ | $\mathrm{mmol\,N\,(mg\,C)}^{-1}$ | $5.0\times10^{-4}$–$1.0\times10^{-2}$ | Minium nitrogen quota |
| 14 | $\phi_N^{(\mathrm{opt})}$ | $1.26\times10^{-2}$ | $\mathrm{mmol\,N\,(mg\,C)}^{-1}$ | $1.0\times10^{-4}$–$5.0\times10^{-2}$ | Optimal nitrogen quota |
| 15 | $\phi_N^{(\mathrm{max})}$ | $1.5\,\phi_N^{(\mathrm{opt})}$ | $\mathrm{mmol\,N\,(mg\,C)}^{-1}$ | 1.0–5.0 | Maximum nitrogen quota |
| 16 | $a_P^{(\mathrm{P})}$ | $2.5\times10^{-3}$ | $\mathrm{m}^3\,(\mathrm{mg\,C})^{-1}\mathrm{d}^{-1}$ | $1.0\times10^{-3}$–$5.0\times10^{-3}$ | Specific affinity constant for phosphorous |
| 17 | $\phi_P^{(\mathrm{min})}$ | $4.29\times10^{-4}$ | $\mathrm{mmol\,P\,(mg\,C)}^{-1}$ | $1.0\times10^{-4}$–$1.0\times10^{-3}$ | Minimum phosphorous quota |
| 18 | $\phi_P^{(\mathrm{opt})}$ | $7.86\times10^{-4}$ | $\mathrm{mmol\,P\,(mg\,C)}^{-1}$ | $1.0\times10^{-4}$–$1.0\times10^{-3}$ | Optimal phosphorous quota |
| 19 | $\phi_P^{(\mathrm{max})}$ | $1.5\phi_P^{(\mathrm{opt})}$ | $\mathrm{mmol\,P\,(mg\,C)}^{-1}$ | 1.0–5.0 | Maximum phosphorous quota |
| 20 | $l_P^{(\mathrm{sink})}$ | 0.75 | - | 0.05–1.0 | Nutrient stress threshold for sinking |
| 21 | $w_P^{(\mathrm{sink})}$ | 0.5 | $\mathrm{m\,d}^{-1}$ | 0.25–1.0 | Maximum sinking velocity |
| 22 | $\alpha_{\mathrm{chl}}^{(0)}$ | $1.52\times10^{-5}$ | $\mathrm{mg\,C\,(mg\,Chl)}^{-1}(\mu\mathrm{E})^{-1}\mathrm{m}^2$ | $5.0\times10^{-6}$–$5.0\times10^{-5}$ | Maximum light utilization coefficient |
| 23 | $\theta_{\mathrm{chl}}^{(0)}$ | 0.016 | $\mathrm{mg\,Chl\,(mg\,C)}^{-1}$ | 0.005–0.05 | Maximum chlorophyll-to-carbon quota |
| | | | **Zooplankton Parameters** | | |
| 24 | $b_Z$ | 0.02 | $\mathrm{d}^{-1}$ | 0.01–0.1 | Basal specific respiration rate |
| 25 | $r_Z^{(0)}$ | 2.0 | $\mathrm{d}^{-1}$ | 1.0–7.5 | Potential specific growth rate |
| 26 | $d_Z^{(0)}$ | 0.25 | $\mathrm{d}^{-1}$ | 0.05–0.5 | Oxygen-dependent specific mortality rate |
| 27 | $d_Z$ | 0.05 | $\mathrm{d}^{-1}$ | 0.025–0.1 | Specific mortality rate |
| 28 | $\eta_Z$ | 0.5 | - | 0.05–1.0 | Assimilation efficiency |
| 29 | $\beta_Z$ | 0.25 | - | 0.05–1.0 | Fraction of activity excretion |
| 30 | $\varepsilon_Z^{(C)}$ | 0.60 | - | 0.05–1.0 | Partition between dissolved and particulate excretion of C |
| 31 | $\varepsilon_Z^{(N)}$ | 0.72 | - | 0.05–1.0 | Partition between dissolved and particulate excretion of N |
| 32 | $\varepsilon_Z^{(P)}$ | 0.832 | - | 0.05–1.0 | Partition between dissolved and particulate excretion of P |
| 33 | $h_Z^{(O)}$ | 0.5 | $\mathrm{mmol\,O_2\,m}^{-3}$ | 0.25–5.0 | Half saturation for zooplankton processes |
| 34 | $h_Z^{(F)}$ | 200.0 | $\mathrm{mg\,C\,m}^{-3}$ | 50.0–500.0 | Michaelis Constant for total food ingestion |
| 35 | $\mu_Z$ | 50.0 | $\mathrm{mg\,C\,m}^{-3}$ | 25.0–100.0 | Feeding Threshold |
| 36 | $\varphi_P^{(\mathrm{opt})}$ | $7.862\times10^{-4}$ | $\mathrm{mmol\,P(mg\,C)}^{-1}$ | $1.0\times10^{-4}$–$1.0\times10^{-3}$ | Optimal phosphorous quota |
| 37 | $\varphi_N^{(\mathrm{opt})}$ | $1.258\times10^{-2}$ | $\mathrm{mmol\,N(mg\,C)}^{-1}$ | $5.0\times10^{-3}$–$5.0\times10^{-2}$ | Optimal nitrogen quota |
| 38 | $\delta_{Z,P}$ | 1.0 | - | 0.05–1.0 | Availability of phytoplankton to zooplankton |
| | | | **Non-living Organic Parameters** | | |
| 39 | $\Lambda_{N3}^{(\mathrm{nit})}$ | 0.01 | $\mathrm{d}^{-1}$ | 0.005–0.1 | Specific nitrification rate at 10°C |
| 40 | $h_N^{(O)}$ | 10.0 | $\mathrm{mmol\,O_2\,m}^{-3}$ | 1.0–10.0 | Half saturation for chemical processes |
| 41 | $\xi_{\mathrm{CO_2}}$ | 0.1 | $\mathrm{d}^{-1}$ | 0.005–0.75 | Specific remineralization rate of particulate carbon |
| 42 | $\xi_{N(1)}$ | 0.1 | $\mathrm{d}^{-1}$ | 0.005–0.75 | Specific remineralization rate of particulate phosphorous |
| 43 | $\xi_{N(3)}$ | 0.1 | $\mathrm{d}^{-1}$ | 0.005–0.75 | Specific remineralization rate of particulate nitrogen |
| 44 | $\zeta_{\mathrm{CO_2}}$ | 0.05 | $\mathrm{d}^{-1}$ | 0.005–0.75 | Specific remineralization rate of dissolved carbon |
| 45 | $\zeta_{N(1)}$ | 0.05 | $\mathrm{d}^{-1}$ | 0.005–0.75 | Specific remineralization rate of dissolved phosphorous |
| 46 | $\zeta_{N(3)}$ | 0.05 | $\mathrm{d}^{-1}$ | 0.005–0.75 | Specific remineralization rate of dissolved nitrogen |
| 47 | $v^{(\mathrm{set})}$ | 1.0 | $\mathrm{m\,d}^{-1}$ | 0.5–10.0 | Settling velocity of particulate detritus |
| | | | **Boundary Condition Parameters** | | |
| 48 | $\lambda_O$ | 0.06 | $\mathrm{m\,d}^{-1}$ | 0.0–0.5 | Relaxation constant for oxygen at bottom |
| 49 | $\lambda_{N(1)}$ | 0.06 | $\mathrm{m\,d}^{-1}$ | 0.0–0.5 | Relaxation constant for phosphate at bottom |
| 50 | $\lambda_{N(2)}$ | 0.06 | $\mathrm{m\,d}^{-1}$ | 0.0–0.5 | Relaxation constant for nitrate at bottom |
| 51 | $\kappa_{N(3)}$ | 0.05 | $\mathrm{m}^2\,\mathrm{s}^{-1}$ | 0.0–0.5 | Relaxation diffusivity for ammonium at bottom |

**Table C2.** List of estimated BFM17 parameters controlling the marine ecosystem dynamics in the model.

| No. | Parameter | Baseline value | Units | BATS | HOTS | Combined |
|---|---|---|---|---|---|---|
| | | | **Phytoplankton Parameters** | | | |
| 1 | $\varepsilon_{\mathrm{PAR}}$ | 0.4 | - | 0.53 | 0.75 | 0.25 |
| 2 | $\lambda_w$ | 0.0435 | $\mathrm{m}^{-1}$ | 0.03 | 0.03 | 0.03 |
| 3 | $c_P$ | 0.03 | $\mathrm{m}^2\,(\mathrm{mg\,Chl})^{-1}$ | 0.005 | 0.005 | 0.005 |
| 4 | $c_{R(2)}$ | $0.1\times10^{-3}$ | $\mathrm{m}^2\,(\mathrm{mg\,C})^{-1}$ | $1.5\times10^{-5}$ | $1.5\times10^{-5}$ | $1.5\times10^{-4}$ |
| 5 | $r_P^{(0)}$ | 1.6 | $\mathrm{d}^{-1}$ | 1.0 | 1.0 | 1.0 |
| 6 | $b_P$ | 0.05 | $\mathrm{d}^{-1}$ | 0.005 | 0.005 | $8.05\times10^{-3}$ |
| 7 | $d_P^{(0)}$ | 0.05 | $\mathrm{d}^{-1}$ | 0.005 | 0.075 | 0.005 |
| 8 | $h_P^{(\mathrm{N,P})}$ | 0.1 | - | 0.04 | 0.015 | 0.005 |
| 9 | $\beta_P$ | 0.05 | - | 0.005 | 0.005 | $9.25\times10^{-2}$ |
| 10 | $\gamma_P$ | 0.05 | - | 0.1 | 0.005 | 0.1 |
| 11 | $a_P^{(\mathrm{N})}$ | 0.025 | $\mathrm{m}^3\,(\mathrm{mg\,C})^{-1}\mathrm{d}^{-1}$ | 0.005 | 0.005 | 0.005 |
| 12 | $h_P^{(\mathrm{N})}$ | 1.5 | $\mathrm{mmol\,NH_4\,m}^{-3}$ | 3.51 | 0.25 | 0.25 |
| 13 | $\phi_N^{(\mathrm{min})}$ | $6.87\times10^{-3}$ | $\mathrm{mmol\,N(mg\,C)}^{-1}$ | $2.62\times10^{-3}$ | $6.24\times10^{-3}$ | $5.74\times10^{-3}$ |
| 14 | $\phi_N^{(\mathrm{opt})}$ | $1.26\times10^{-2}$ | $\mathrm{mmol\,N(mg\,C)}^{-1}$ | $1.61\times10^{-3}$ | $7.3\times10^{-3}$ | $5.77\times10^{-3}$ |
| 15 | $\phi_N^{(\mathrm{max})}$ | $1.5\phi_N^{(\mathrm{opt})}$ | $\mathrm{mmol\,N(mg\,C)}^{-1}$ | 1.44 | 1.0 | 1.0 |
| 16 | $a_P^{(\mathrm{P})}$ | $2.5\times10^{-3}$ | $\mathrm{m}^3\,(\mathrm{mg\,C})^{-1}\mathrm{d}^{-1}$ | 0.005 | $1.89\times10^{-3}$ | 0.005 |
| 17 | $\phi_P^{(\mathrm{min})}$ | $4.29\times10^{-4}$ | $\mathrm{mmol\,P(mg\,C)}^{-1}$ | $1.0\times10^{-4}$ | 0.001 | $2.01\times10^{-4}$ |
| 18 | $\phi_P^{(\mathrm{opt})}$ | $7.86\times10^{-4}$ | $\mathrm{mmol\,P(mg\,C)}^{-1}$ | $1.09\times10^{-4}$ | $1.69\times10^{-4}$ | $1.80\times10^{-4}$ |
| 19 | $\phi_P^{(\mathrm{max})}$ | $1.5\phi_P^{(\mathrm{opt})}$ | $\mathrm{mmol\,P(mg\,C)}^{-1}$ | 1.0 | 5.0 | 1.0 |
| 20 | $l_P^{(\mathrm{sink})}$ | 0.75 | - | 0.05 | 0.05 | 0.21 |
| 21 | $w_P^{(\mathrm{sink})}$ | 0.5 | $\mathrm{m\,d}^{-1}$ | 1.0 | 0.25 | 0.25 |
| 22 | $\alpha_{\mathrm{chl}}^{(0)}$ | $1.52\times10^{-5}$ | $\mathrm{mg\,C(mg\,Chl)}^{-1}\mathrm{\mu E}^{-1}\mathrm{m}^2$ | $7.31\times10^{-6}$ | $5.0\times10^{-5}$ | $7.32\times10^{-6}$ |
| 23 | $\theta_{\mathrm{chl}}^{(0)}$ | 0.016 | $\mathrm{mg\,Chl(mg\,C)}^{-1}$ | 0.005 | $4.57\times10^{-2}$ | 0.005 |
| | | | **Zooplankton Parameters** | | | |
| 24 | $b_Z$ | 0.02 | $\mathrm{d}^{-1}$ | 0.01 | 0.01 | 0.01 |
| 25 | $r_Z^{(0)}$ | 2.0 | $\mathrm{d}^{-1}$ | 4.08 | 7.5 | 1.5 |
| 26 | $d_Z^{(0)}$ | 0.25 | $\mathrm{d}^{-1}$ | 0.5 | 0.5 | 0.3 |
| 27 | $d_Z$ | 0.05 | $\mathrm{d}^{-1}$ | 0.1 | 0.1 | 0.1 |
| 28 | $\eta_Z$ | 0.5 | - | 0.58 | 1.0 | 1.0 |
| 29 | $\beta_Z$ | 0.25 | - | 0.87 | 0.40 | 0.76 |
| 30 | $\varepsilon_Z^{(C)}$ | 0.60 | - | 0.05 | 0.05 | 0.05 |
| 31 | $\varepsilon_Z^{(N)}$ | 0.72 | - | 1.0 | 0.05 | 1.0 |
| 32 | $\varepsilon_Z^{(P)}$ | 0.832 | - | 0.9 | 1.0 | 0.87 |
| 33 | $h_Z^{(O)}$ | 0.5 | $\mathrm{mmol\,O_2\,m}^{-3}$ | 5.0 | 0.25 | 0.48 |
| 34 | $h_Z^{(F)}$ | 200.0 | $\mathrm{mg\,C\,m}^{-3}$ | 500.0 | 500.0 | 500.0 |
| 35 | $\mu_Z$ | 50.0 | $\mathrm{mg\,C\,m}^{-3}$ | 100.0 | 100.0 | 74.5 |
| 36 | $\varphi_P^{(\mathrm{opt})}$ | $7.862\times10^{-4}$ | $\mathrm{mmol\,P(mg\,C)}^{-1}$ | $3.12\times10^{-4}$ | $1.0\times10^{-3}$ | $6.93\times10^{-4}$ |
| 37 | $\varphi_N^{(\mathrm{opt})}$ | $1.258\times10^{-2}$ | $\mathrm{mmol\,N(mg\,C)}^{-1}$ | $4.61\times10^{-2}$ | $4.66\times10^{-2}$ | $4.85\times10^{-2}$ |
| 38 | $\delta_{Z,P}$ | 1.0 | - | 0.16 | 1.0 | 1.0 |
| | | | **Non-living Organic Parameters** | | | |
| 39 | $\Lambda_{N3}^{(\mathrm{nit})}$ | 0.01 | $\mathrm{d}^{-1}$ | $1.03\times10^{-2}$ | 0.1 | 0.1 |
| 40 | $h_N^{(O)}$ | 10.0 | $\mathrm{mmol\,O_2\,m}^{-3}$ | 1.0 | 2.67 | 4.35 |
| 41 | $\xi_{\mathrm{CO_2}}$ | 0.1 | $\mathrm{d}^{-1}$ | 0.005 | 0.005 | 0.005 |
| 42 | $\xi_{N(1)}$ | 0.1 | $\mathrm{d}^{-1}$ | $1.58\times10^{-2}$ | 0.75 | 0.005 |
| 43 | $\xi_{N(3)}$ | 0.1 | $\mathrm{d}^{-1}$ | 0.75 | 0.75 | 0.75 |
| 44 | $\zeta_{\mathrm{CO_2}}$ | 0.05 | $\mathrm{d}^{-1}$ | 0.75 | 0.75 | $4.1\times10^{-2}$ |
| 45 | $\zeta_{N(1)}$ | 0.05 | $\mathrm{d}^{-1}$ | 0.42 | 0.75 | 0.75 |
| 46 | $\zeta_{N(3)}$ | 0.05 | $\mathrm{d}^{-1}$ | 0.005 | 0.75 | $3.38\times10^{-2}$ |
| 47 | $v^{(\mathrm{set})}$ | 1.0 | $\mathrm{m\,d}^{-1}$ | 10.0 | 0.5 | 10.0 |
| | | | **Boundary Condition Parameters** | | | |
| 48 | $\lambda_O$ | 0.06 | $\mathrm{m\,d}^{-1}$ | 0.5 | 0.5 | 0.5 |
| 49 | $\lambda_{N(1)}$ | 0.06 | $\mathrm{m\,d}^{-1}$ | 0.13 | 0.5 | 0.17 |
| 50 | $\lambda_{N(2)}$ | 0.06 | $\mathrm{m\,d}^{-1}$ | 0.12 | $3.24\times10^{-6}$ | $3.74\times10^{-2}$ |
| 51 | $\kappa_{N(3)}$ | 0.05 | $\mathrm{m}^2\,\mathrm{s}^{-1}$ | $2.51\times10^{-5}$ | $1.20\times10^{-4}$ | $1.25\times10^{-4}$ |

*Code and data availability.* All codes and data necessary to reproduce the results in this paper have been archived on Zenodo. The parameter estimation is performed by coupling the BFM17+POM1D code (Smith et al., 2020) to DAKOTA the code, which is archived in Kern et al. (2023a). The code used to perform the one-at-a-time sensitivity study of the model parameters is archived at Kern et al. (2023b). All data and plotting scripts required to generate the figures included in this paper are archived at Kern et al. (2023c).

*Author contributions.* SK, KMS, NP, and PEH developed the optimization-based parameter estimation method, MAM developed the HOTS test case, KEN provided input on the numerical optimization approach, NLS and NP provided guidance on data sources and the physical interpretation of results, SK performed all parameter estimations and produced all results presented in the paper, SK and PEH prepared the initial draft of the paper, and all authors edited the paper to produce the final version.

*Competing interests.* The authors declare that they have no conflict of interest.

*Acknowledgements.* SK was supported by an ANSEP Alaska Grown Fellowship and by a National Science Foundation (NSF( Graduate Research Fellowship Program award. This material is based upon work supported by the NSF under grants OCE-1924636 and OCE-1924658. The authors would also like to acknowledge high-performance computing support from Cheyenne (doi:10.5065/D6RX99HX) provided by NCAR's Computational and Information Systems Laboratory, sponsored by NSF.

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
