# Peer review of "Computationally efficient parameter estimation for high-dimensional ocean biogeochemical models"

_Geoscientific Model Development, 2023_

## Author Comment (AC1)

Response to Reviewer #1
We greatly appreciate the time taken by the reviewer to read our manuscript. We have taken into consideration and addressed all comments, questions, and suggestions from the reviewer, and we feel that the revised manuscript is now substantially stronger as a result. Changes made to the text at the request of the reviewer have been highlighted in red in the revised manuscript. In the following, reviewer comments are repeated in blue italics and our responses are provided in the bulleted sections of text.

*The article presents a hybrid optimization method which is implemented using the optimization toolkit DAKOTA. The method is applied in order to optimize the parameters of the recent BGC model configuration BFM17 developed by some of the papers' authors (Smith et al., 2021).*

*The proposed method is a combination of (many) Latin hypercube samples (LHS) of the parameter space and a gradient-based optimization (gradient search, (GS)) using some best-ranked LHS samples as initial search points.*

*Although both, LHS and GS have been examined before in combination with population-based search methods like genetic algorithms (GAs) the proposed combination of LHS and GS seems to be new.*

*The application to the BFM17 BGC model is validated in a twin experiment setup where it is able to detect at least the misfit-sensitive parameters. A drawback of the twin experiment is that only a time window of 30 days is considered (according to line 294 of the paper) which does not represent the full annual cycle.*

- In response to this helpful comment, as well as a similar comment from the other reviewer, we have performed an additional Twin Simulation Experiment (TSE) that more closely matches the annual, five-field optimization that is the focus of the parameter estimation at the BATS and HOTS locations. The results of this TSE are included as a new Figure 6 in the revised paper, with substantial additional text included at the end of Section 4.1 to discuss the results of this TSE. Briefly, we found in the annual TSE that more parameters had a relative importance greater than 0.01, as compared to the 30-day tests. This is indicative of a more complex optimization problem where many more parameters can affect the results. Consequently, although we recover the baseline parameter values across the range of relative importance values, there are still some parameters that we do not fully recover. However, most parameter values that do not reach the baseline value do at least approach the value. As we now discuss at greater length at the end of Section 4.1, the comparison of the 30-day and annual TSE results demonstrates the challenge of estimating many sensitive parameters in a complex objective function space, even when using a gradient based approach.

*A calibration of all model parameters against 1D observations yields a new and justified parameter set for BFM17 (only a manually tuned parameter set has been provided before):*

*An RMSD type model-data misfit w.r.t. five tracers does significantly improve in comparison to the manually determined parameter set and, consequently, model simulation results do better agree with their observational counterparts (average mismatches drop from several standard deviations to less than one standard deviation).*

*However, even more convincing would be a calibration of BFM17 coupled to a global circulation model, and against global 3D observations, since (according to Smith et al., 2021) BFM17 is designated to be used in global 3D model configurations.*

- We completely agree that this would be the ideal approach. However, even with the cost savings enabled by a smaller BGC model such as BFM17, global circulation models (GCMs) would still be extremely expensive to evaluate many tens of thousands of times, as is required even when using a gradient-based approach. It is common in optimization approaches to use surrogate or lower-fidelity models to accelerate the optimization process, even when the intended application of the optimized parameters is a higher-fidelity simulation. In this sense, the current approach effectively employs

POM1D as a physics-based, low-cost replacement for a GCM. Because this is such an important point, we now re-emphasize this point in the final paragraph of the conclusions section of the paper.

*It would also be interesting to see, how a population-based search method (alternatively incorporating LHS and GS) would perform in comparison to the proposed method. I notice that the computational demand of 25,000 function evaluations for LHS would be comparable with many iterations of, e.g., an evolutionary algorithm.*

- We agree that a genetic or evolutionary algorithm could be a promising alternative strategy to the optimization problem. We did in fact consider such an approach and estimated the smallest possible population size to be roughly 360. With such a large population size, the number of iterations required to meet reasonable convergence criteria was prohibitive; initial testing with this approach did not produce converged solutions. By contrast, a major advantage of the initial LHS step in the present method is that we can run many model evaluations in an embarrassingly parallel fashion. So, while the computational cost of running many LHS samples may be comparable to running a genetic algorithm (GA) to convergence, the real time taken to perform the optimization would be much less since there are limits to what can be parallelized when implementing a GA. It is conceivable that truncated GAs could be used instead of the LHS, and we have mentioned this approach in the last paragraph of Section 2.2. To address the broader point from the reviewer, we have also added a paragraph to the Conclusions.

*Summarizing, I suggest a minor revision to mention hybrid population-based search methods using LHS on the one hand and GS on the other hand. Also, though the main focus of the paper is on presenting and testing the proposed optimization method and providing a suitable parameter set for BFM17, I wonder if the calibration experiments reveal some general relationships between parameter values and model skills, one could elaborate on?*

- As outlined in our response to the previous reviewer comment, we agree that it is important to mention hybrid population-based approaches and have added text to this effect in Section 2.2 and the Conclusions. We also agree that it would be interesting to determine if there are general relationships between parameter values. As mentioned by the reviewer, the primary focus of this paper was on the calibration and, due to the number of parameters we are optimizing simultaneously, it is difficult to succinctly interpret the calibrated values for the full set of 51 parameters. Nevertheless, we have added some text in Section 4.3 discussing the relationship between two specific parameters, as an example of what this type of analysis would look like.

*Some technical corrections:*
*- The DOI link of Vichy et al., 1998, does not work in the paper PDF.*

- Thank you for finding this error. The link is split across two lines, which is why it does not work. We will ensure that the link is on one line only in the final manuscript (we were unfortunately not able to control this in the revised paper).

*- Line 176: "A detailed description ... has been outlined in detail ..." (2 x "detail", drop one)*

- This has now been corrected.

*- Line 305 ff: the parameter disturbance seems to be 20% instead of the stated 10%*

- In the initial twin simulation experiment, the 10% perturbation is from the nominal value. The range of values is then set to +/- 25% of the nominal value. So, the perturbation will not have a 0.1 difference initially, because the normalization is not based exactly on the nominal value but the upper and lower bounds. In the initial TSE experiment we get a constant 0.2 shift because the parameter range is 50% of the nominal parameter value as we are using +/- 25% of the nominal value (10% of 1 is 20% of 0.5). In the additional TSE, the bounds are not a percentage of the nominal parameter value so the initial displacement is not a constant proportion of the parameter range.

*- Line 305: The definition of S^ is difficult to read. Perhaps replace "sensitivity metric S(pi)" by "sensitivity factor S(pi)" and explicitly write "S^(pi)=S(pi)/Smax with Smax=max_i(S(pi))"*

- Thank you for this recommendation. Based on this comment and one from the other reviewer, we have added Eq. (6) to make the calculation of the sensitivity factor clearer, as well as updated the text to add clarity.

*- Lines 299 and 303 and Table 1: what does "nominal parameter value" mean? Is it the "baseline parameter value" from the manual calibration by Smith et al.?*

- Yes, this is correct, the nominal parameters are the baseline parameter values taken from Smith et al. The wording has been changed to consistently use baseline to avoid ambiguity.

*- Figure 5: what is pi^ and po^? Is it the parameter values, normalized w.r.t. their credible interval length?*

- This is correct, $\hat{p}_i$ is the normalized parameter value being tested by the optimization algorithm, while $\hat{p}_o$ is the normalized baseline parameter value. The parameters are normalized between 0 and 1 with respect to their upper and lower bounds. We have added a comment to the caption of the relevant figures clarifying these parameters and to the main text. The clarification is included with the addition to address the concern with respect to Line 305 above.

*- Table 1: normalized RMSD values? It is according to Equation (2), i guess?*

- This is correct, the normalized RMSD values are calculated according to Eq. (2). The table caption now specifies this to avoid ambiguity.

*- Line 364: "year round" => "all year round"*

- This has now been corrected.

*- Line 366: "maxima" => "maximum"*

- This has now been corrected.

We thank the reviewer for these useful comments, and the paper has now been revised to address all the above points. Sincerely, the authors.

---

## Author Comment (AC2)

Response to Reviewer #2

We greatly appreciate the time taken by the reviewer to read our manuscript. We have taken into consideration and addressed all comments, questions, and suggestions from the reviewer, and we feel that the revised manuscript is now substantially stronger as a result. Changes made to the text at the request of the reviewer have been highlighted in red in the revised manuscript. In the following, reviewer comments are repeated in blue italics and our responses are provided in the bulleted sections of text.

*The authors present a hybrid parameter estimation technique which includes a gradient-free first optimization stage and a gradient-based second stage. The two-stage approach applied to a 1D coupled physical-biogeochemical ocean model appears to work well, but for a study focused on methodology, more details about it should be included.*

**General comments**

*The study makes a relatively straightforward case, the text in general and the description of the methods is mostly easy to follow, and the experiments are well motivated (maybe except for the TSE, see below).*

*However, the manuscript frames this study as one primarily focused on the demonstration of the parameter estimation method. And here, I think, more emphasis could be placed on the parameter estimation part.*

*(1) As a reader, it would be good to get a better idea of the computational cost in terms of runtime or number of model/function evaluations, in particular comparing the initial global search to the gradient-based second stage.*

- We agree that it is important to clarify the computational expense of the proposed method, particularly since many of the decisions made were geared toward increasing computational efficiency. To address this point, we have now included a new Appendix B in the revised paper that outlines the cost of running each step in the methodology. In brief, the sampling computational time was 12,571, 12,247, and 45,000 cpu-hours for BATS, HOTS, and the multi-site calibration cases, respectively. The total computational time was 31,052, 21,279, and 81,951 cpu-hours, with 60%, 42%, and 45% of the compute time spent on the optimization portion of the two-step process.

*(2) It seems sensible to use the two-stage (global+local) parameter estimation, but how does each stage contribute to the decrease in the cost function? Readers may ask if after 25000 simulations in stage 1, is there any need for stage 2? Or, how well does stage 2 do if started from the baseline?*

- This is another great point and, in addition to exploring the computational cost of each stage in the optimization methodology, we have included more detail in Appendix B of the relative improvements in model agreement from the random sampling and the local optimization portions of the method. As shown in Figure B1, in all cases we can further reduce the error after the initial global step by doing local optimization. For BATS, HOTS, and the multi-site cases the error decreases by 31%, 91%, and 85% respectively. Note that most of the improvements are for HOTS which is initially just very bad.

*(3) Studies like this one often include replicate experiments, why not include one here, to show that similar parameter values are recovered when running the estimation again.*

- We agree that replicate experiments can be an important aspect of parameter estimation studies and text near the beginning of Section 4.2 noting this point. We have chosen not to do such experiments in the present study because of the random nature of the initial global search in the 51-dimension parameter space considered here. That is, with only 25,000 samples in the Latin hypercube step, we will likely start the second-phase gradient-based optimizations in the replicate experiments from a completely unique set of parameter values, resulting in different final parameter values. Based on our tests, however, the 25,000 samples in the initial global search are sufficient to ensure that each successive application of the overall method will result in better agreement with the observational data than the baseline values from Smith et al. – it is just the extent of the improvement that may differ.

*(4) 25000 model evaluations are a lot, which basically prohibits the use of 3D models (BGC model coupled to 3-dimensional circulation models). What could be done to reduce runtime, would the authors consider the use of a more sophisticated gradient-free method (some are described in the introduction) instead of Latin hypercube sampling?*

- We thank the reviewer for drawing attention to these important points. Regarding ways to reduce the runtime, there are two distinct ways to reduce the overall computational cost of performing the parameter estimation. The first is to cheapen the cost of running the model, while the second is to reduce the number of model evaluations performed. For the former, we cannot significantly reduce the cost of evaluating BFM17+POM1D, which is already effectively a physics-based surrogate for more detailed dynamical models. This is now discussed in the final paragraph of the Conclusion. To reduce the overall number of model evaluations, it is indeed possible that other gradient-free methods, such as genetic algorithms (GAs) could be used. However, for the present study, we instead chose Latin hypercube sampling (LHS) because many model evaluations can be easily performed in an embarrassingly parallel fashion. So, while the computational cost of running many LHS samples may be comparable to (or even slightly greater than) running a GA to convergence, the real time taken to perform the optimization is much less since there are limits to what can be parallelized when implementing a genetic algorithm. It is conceivable that truncated GAs could be used instead of the LHS, and we have mentioned this approach in the last paragraph of Section 2.2. To address the broader point from the reviewer, we have also added a new fourth paragraph to the Conclusions.

*In the twin simulation experiment (TSE; section 4.1), it reads like all 17 state variables were used as synthetic observations. But not all 17 variables are part of the BATS or HOTS datasets, in fact, only 5 appear to be used. How do the results of the TSE change if only 5 state variables are observed? This would appear to be a much more important experiment than one using all the variables. Additionally, a little later in the section (l 312): "While these results may suggest that the least sensitive parameters could be excluded from the subsequent calibration studies, redoing the sensitivity study with our standard objective function reveals larger relative importance values for the full set of parameters.": This is a bit of an unsatisfactory result, could the discrepancy be due to including 5 compared to 17 observation types in the objective function?*

- In response to this helpful comment, as well as a similar comment from the other reviewer, we have performed an additional twin simulation experiment (TSE) that more closely matches the annual, five-field optimization that is the focus of the parameter estimation at the BATS and HOTS locations. The results of this TSE are included as a new Figure 6 in the revised paper, with substantial additional text included at the end of Section 4.1 to discuss the results of this TSE. Briefly, we found in the annual TSE that more parameters had a relative importance greater than 0.01, as compared to the 30-day tests. This is indicative of a more complex optimization problem where many more parameters can affect the results. Consequently, although we recover the baseline parameter values across the range of relative importance values, there are still some parameters that we do not fully recover. However, most parameter values that do not reach the baseline value do at least approach the value. As we now discuss at greater length at the end of Section 4.1, the comparison of the 30-day and annual TSE results demonstrates the challenge of estimating many sensitive parameters in a complex objective function space, even when using a gradient-based approach. Concerning the last question from the reviewer, the discrepancy in sensitivities comes from having a different objective function. In one sense it is not a discrepancy and is instead the result of the problem being fundamentally different. However, in terms of making sure that the optimizer is decreasing the error as expected and demonstrating that DAKOTA and our model are interfacing correctly, we feel that the present TSEs are sufficient.

*Specific comments*
*L 72: "in a high-dimensional BGC model across a range of ocean conditions, using a 1D model for vertical ocean mixing": Here it would be nice for the reader to explain a bit better what is meant by dimensionality: "high-dimensional BGC model" and "1D model for vertical ocean mixing" are used together and at this*

*point it is unclear if "high-dimensional" refers to 3 spatial dimensions (represented at various spatial locations as 1D vertical models) or if it is referring to the dimensions of the state space. More generally, it would be useful to describe what is to follow in a bit more detail early on. Even the formulation in line 5 "simultaneous parameter estimation at multiple ocean locations" is a bit ambiguous, and the kind of model setup (1D, 3D) that is being used in the study could be mentioned earlier.*

- We appreciate this suggestion and have now attempted in the abstract, introduction, and several other places in the paper to be more specific when referring to "dimensionality".

*L 98: I presume \Pi is a matrix of weights and \Pi_{i,j} is a scalar weight?*

- Yes, Pi is a matrix of weights, while Pi_{i,j} are scalar weights for the corresponding field at the corresponding ocean site. In this work, we use an identity matrix for Pi but include it in the generalized description of the methodology to emphasize that weights can be added. We point out that the weights can be formulated differently in the description of the method and in the conclusion, and the corresponding line noted by the reviewer has been updated to improve clarity.

*L 99: "describes the misfit with observational (or other reference) data": For clarity, I suggest adding "of the model output" or similar.*

- The corresponding line has been updated for clarity.

*Eq 2: I would have expected that the 1/\sigma term (a weight) would be contained in \Pi. In fact, the subsection makes no further reference to \Pi, and it would be useful to explain here what it is used for.*

- We thank the reviewer for bringing this potential point of confusion to our attention. We consider the 1/sigma term to be a normalizing factor in the error function (Eq. 2) and different forms of this function, with different normalizations, are considered in Appendix A2. The weights Pi are included here for generality and could be used to provide more emphasis on certain observational fields, for example those that are deemed to be of greater importance or for which uncertainties are lower. We have now added text in Section 2.1 after Eq. (2) to explain both these points.

*Fig. 1: It is somewhat easy to trace the line from model run and comparison with data to "Calculate model error", feeding it into DAKOTA and obtaining new parameters as output with which to run the model again. but what do the dotted gray lines mean and why does the calculated model error bypass the interface? A minor complaint is that terms like BFM17 and \partial A_j/ \partial t have not been mentioned in the text when this figure is first referenced.*

- To address the concerns, the caption for Figure 1 has been updated. The grey dotted lines are intended to show how the different components are implemented in practice – the error calculation output is formatted so that DAKOTA can read it, so it does not have to be interpreted by the interface.

*L 151: "how quickly the values of J increase when the N_random simulations are sorted": That sounds a bit like J increases while the simulations are being sorted, to avoid confusion maybe use something like: "how quickly the values of J increase in the sorted N_random simulations".*

- The proposed change was made for clarity.

*L 164: "The QN algorithm reliably and efficiently converged to optimized solutions.": Is this a result from this study or a general observation? Maybe add "In our experiments ...".*

- This change has been made.

*L 165: "Similar to the ecosystem parameter estimation study by Matear (1995), we found that the conjugate gradient method failed to converge efficiently." It reads as if this is still meant as a justification for using QN over the conjugate gradient method. I suggest rephrasing it slightly: "In comparison, we found that the*

*conjugate gradient method failed to converge efficiently, a result similar to that in the ecosystem parameter estimation study by Matear (1995)."*

- We agree with the reviewer that this point could be clearer and have made the suggested change.

*L 192: "Each CFF is represented as a vector where each element is a constituent component concentration corresponding to a state variable.": Initially, I thought that this meant that the state variables are divided up into multiple vectors according to their type/function, which is not the case according to the following paragraph. This could be explained better without adding much more text, maybe "Each CFF is a vector representing a model variable divided into the elemental constituents represented by the model, e.g., the phytoplankton CFF is a 4-element vector containing the C, N, P and chlorophyll concentration of the phytoplankton variable (see below for more details)."*

- The text has been edited to add clarity.

*L 193: "BFM17 was simplified to be a general, but computationally cheaper, model that retains the essential BGC processes for modeling a phytoplankton spring bloom ...": Is this mean in comparison to BFM56?*

- Yes, this statement was made for BFM56. The text has been edited to add clarity.

*L 232: At this point, it is unclear from the text what parameter values are used. Interestingly, this information, plus comments about the manually adjusted parameters, is provided in the description of Fig. 4 in the next subsection, which is helpful to the reader. I would suggest moving the information that applies to both datasets (use of baseline model parameters, single-site model calibration and the multi-site calibration etc.) to the description of Fig. 3.*

- A reference to the origin of the baseline parameter values is now made in the earlier subsection, with the later subsection being edited accordingly.

*L 303: "nominal value": Does this refer to the optimized value?*

- Nominal value refers to the baseline or target parameter values we are trying to recover over the course of the twin simulation experiment. The verbiage in the paper has been updated for clarity.

*L 304: So some parameters were only perturbed down, because perturbing them up would hit the baseline + 25% boundary? What then makes up the second perturbation case, the optimized value? This could be explained better.*

- The sensitivity analysis was performed by perturbing values by 5% except when this would exceed the standard bounds (i.e., those in Table C1), not the alternative +/- 25% bounds. The text has been updated to clarify this point.

*L 305: "... defined as the maximum objective function evaluation between the two perturbation cases for each parameter." This sentence is difficult to understand, I suggest rephrasing it.*

- Based on this comment as well as one from the other review, we have added Eq. (6) to make the calculation of the sensitivity factor clearer. We have also added corresponding text around this equation.

*L 317: It would be useful to state what $N_v$ is set to here. I presume it is 5, based on previous figures. But it is not stated directly in the text.*

- Yes, we have used $N_v = 5$, corresponding to the five target observational fields. This is now explicitly addressed at the beginning of Section 4.2.

*L 351: "improved by a factor of ... over 236 (for nitrate)": True, but this large improvement is more a function of the enormous misfit in the baseline experiment.*

- This is indeed true, and we have added text at the beginning of Section 4.2.2 noting the enormous initial error at the HOTS location.

We thank the reviewer for these useful comments, and the paper has now been revised to address all the above points. Sincerely, the authors.

---

## Author Response (AR2)

Response to Reviewer #2
We greatly appreciate the time taken by the reviewer to read our manuscript. We have taken into consideration and addressed all comments, questions, and suggestions from the reviewer, and we feel that the revised manuscript is now substantially stronger as a result. Additional changes made to the text at the request of the reviewer have been highlighted in red in the newly revised manuscript. In the following, reviewer comments are repeated in blue italics and our responses are provided in the bulleted sections.

*The authors have mostly addressed my comments, and I am happy about the addition of appendix B. However, results from the additional twin simulation experiment (TSE) that was conducted, highlight differences between parameter estimation experiments including 5 compared to 17 observation types. Yet, this aspect is not examined adequately in the manuscript.*

*# general comments*

*The introduction provides a nice overview of recent parameter estimation studies for BGC ocean models. However, it focuses mostly on the computational difficulties associated with the estimation of a large number of uncertain parameters. Other big problems that are not mentioned are that of parameter dependency and limited data availability (not in the spatial or temporal sense, but in the type of data commonly observed). For example, low phytoplankton growth and mortality rates may yield similar phytoplankton model estimates to high growth and mortality rates -- the values of phytoplankton growth and mortality rates are difficult to determine with phytoplankton data alone. In terms of data availability, chlorophyll data (the most commonly used data type for BGC data assimilation) may be useful for constraining phytoplankton parameters -- with the caveats just mentioned above -- but almost useless in estimating a parameter related to detritus, e.g., a decomposition rate. Both of these problems are more likely to occur in BGC models with large numbers of uncertain parameters. In fact, they may explain some of the differences seen in the TSE in which all state variables are observed (the 30-day TSE) compared to that in which only 5 variables are observed (the annual TSE). Fewer data types constrain fewer parameters and, since the objective function is based on fewer data types, sensitivity values can differ greatly.*

*Here, it is unfortunate that the two TSEs in question differ in two aspects (number of observed variables/data types and length of the observations), so it becomes difficult to estimate to what degree each aspect causes the difference in the results. I would suggest a new experiment in which only one of the two aspects is changed from the 30-day TSE, but I can understand if this is not possible due to computational constraints.*

*Based on my comment above, and even without additional experiments, the authors should mention and discuss parameter dependency and limited data availability as important parameter estimation problems in the introduction, which is currently very much focused on computational issues.*

*Furthermore, the difference in the number of observed variables should be emphasized more in section 4.1 when interpreting the results of the TSEs. The expanded interpretation of results may also warrant a new paragraph in the Conclusions section.*

We thank the reviewer for their continued valuable comments, which we summarize here: (i) Additional consideration should be paid to other challenges that limit our ability to predict parameter values; and (ii) The two TSEs presented in the paper change multiple aspects simultaneously.

Regarding the first point, we have added a new paragraph near the beginning of the Introduction that addresses the issues of data sparsity and parameter dependence raised by the reviewer, which are indeed valid and pressing concerns. This additional text has also allowed us to restate the purpose of this paper more clearly for the reader. That is, we are primarily focused on presenting and demonstrating a novel calibration methodology that is computationally efficient for large sets of parameters. This methodology is sufficiently flexible that different models with fewer parameter dependencies and

different reference data that is less sparse can be examined in future studies. These points are additionally reiterated in Section 4.1 in the revised paper where we discuss the results of the TSEs.

To address the reviewer's second point, we have run two additional TSEs that use all 17 state variables in the objective function with the full parameter bounds from Table C1. In the first case we simulated 30 days only, like the TSE shown in Fig. 5 where we used tighter parameter bounds. In the second, we calculated monthly averages for the last year of three-year model runs, such that the resulting TSE can be compared to the results in Fig. 6 where only five state variables were used in the objective function. The resulting recovery of baseline parameter values for the two additional TSEs is qualitatively like those shown in Figs. 5 and 6, respectively. Because of this similarity, we do not show these additional TSE results in the revised paper, but these results are now discussed in a new paragraph near the end of Section 4.1. In particular, the additional TSEs suggests that the duration of the model runs, more so than the parameter bounds or number of state variables, may be responsible for much of the difference between Figs. 5 and 6. We also reiterate the possibility that the previously discussed issues of data sparsity and parameter dependence may contribute to a decrease in the number of fully recovered parameters.

*# specific comments (line numbers correspond to tracked-changes document)*

*L 4: "ocean high-dimensional (in parameter space) BGC models" → "high-dimensional (in parameter space) BGC ocean models"*
- The recommended change has been made.

*L 9: "objective functions": some readers may be more familiar with the terms "cost function" or "loss function" and I would suggest adding a brief description (such as "which quantifies the error between observations and the model estimate") to make the abstract more accessible.*
- These changes have been made to improve clarity.

*L 27: "Subsequent to verification using the TSE, we use the method to estimate parameters for the two sites, both individually and together.": Mention here that "real" data are used for this.*
- We have clarified this point as requested at the end of the third paragraph in the Introduction of the revised paper.

*L 102: "these weights can be adjusted as desired, for example [...] to provide greater weight to state variables for which observational uncertainties are smaller": While true, this statement makes it appear as if this objective function does not provide greater weight to state variables for which observational uncertainties are smaller, when in fact it does so via the \sigmas. I would suggest changing the statement to "these weights can be adjusted as desired, for example [...] to provide greater weight to select ocean sites".*
- This is an interesting point and touches on the differences between observational uncertainties in the data and physical variability in the data (which are both often represented using the symbol sigma). To clarify our intentions, we have made the sentence reference by the reviewer more general and provided a comment after Eq. (2) specifying that our sigma represents the latter type of variability referenced above.

*Fig 1. caption: "calculates the error": I would suggest using "calculates the current value of the objective function" just because the diagram shows the "Objective Function Calculator" and no mention of "error".*
- The caption has been updated accordingly.

*Fig 5. caption: Mention what S-hat is. Consider adding a shaded area or dotted blue lines to indicate the +/-5% range around the baseline value, indicating that a parameter is considered "recovered".*

- The captions for Figures 5 and 6 have been updated to describe the meaning of S-hat and refer the reader to Eq. (6) where it is defined. We have also added horizontal dashed blue lines in Figures 5 indicating +/-5% around the baseline value, as suggested by the reviewer. We have not done the same to Figure 6 since the parameter range is not a constant percentage of the nominal value so the +/- 5% is not a constant normalized distance from the baseline values for all parameters, and therefore cannot cleanly be represented using single continuous lines.

*Fig 5. caption: Consider using "p-hat_i - p-hat_O, the difference between the test and baseline normalized parameter values" instead of "the difference between the test and baseline normalized parameter values, p-hat_i and p-hat_O, respectively", it directs the reader's eyes directly to "p-hat_i - p-hat_O" on the right y-axis.*
- This change has been made in the captions for Figures 5 and 6.

*Fig 5. caption: To me personally, the sentence "The parameter values are normalized between 0 and 1 based on the upper and lower bounds for each parameter." is not helpful here.*
- We have removed this sentence from the captions of Figures 5 and 6.

*L 230: "Five parameters fell into this category": Due to changes in the text preceding this statement, it is no longer clear what "this category" is referring to, I would suggest rephrasing it.*
- We now clarify that we are referring here to the fact that five parameters are perturbed in only one direction.

*L 231: "maximum objective function evaluation for each parameter between the positive and negative 5% perturbation cases": The "between" makes it sound like all parameter values in the +/- 5% range were tested, which I do not think is the case. I would further suggest using "value" instead of "evaluation".*
- We have now updated this sentence to make it clear that we are referring to the maximum of the two objective function values corresponding to the 5% positive and negative perturbations.

*L 480: "This shows that the water is clearer than initially estimated.": I would be a bit more careful in the wording when interpreting estimated parameter values; my suggestion: "This result suggests that the water is clearer than initially estimated."*
- Thank you for pointing this out and we have updated the wording as suggested by the reviewer.

*L 636: "The run time for a single model evaluation is approximately 5 min.": Run on a single node, single core, or what configuration?*
- The text has been updated to indicate that the cost of running the model is 5 min on a single core.

*Fig. B1: The apparent lack of correlation between initial sampling and final objective function values may suggest that it is not important to start the gradient-based optimization at a low value, but rather to start the optimization at different locations in parameter space to avoid local minima. But any additional experiments would be beyond the scope of this study.*
- The reviewer makes a very good point, since we are subject to the local nature of the gradient based method. Falling into local minima is a primary concern which suggests that it may be worth considering relative locations of the initial parameter sets for the optimization runs. The improvement in the baseline cases emphasizes the fact that we cannot rule out the possibility that a parameter set with a higher initial error but in a different region of the parameter space could result in better agreement than that achieved by the 20 best randomly sampled parameter sets in our results. Using the best cases was assumed to be the best approach for identifying regions of relatively low error to proceed with an optimization runs.

  That being said, this methodology could be modified to use a different criterion for selecting the initial parameter sets. For example, instead of only using the error based objective function, one could

incorporate some measure of distance to the selection criteria to ensure that we are getting reasonable coverage of the parameter space. As mentioned, this was not done in this work but it is worth highlighting and should be considered in the future. The major issue that will be faced in taking on this will be determining a way to ensure sufficient coverage of the parameter space for such a high-dimensional parameter space, since this is what motivated the random sampling in the first place. This a discussion of this important point raised by the reviewer has been added to Appendix B.

We thank the reviewer for these useful comments, and the paper has now been revised to address all the above points. Sincerely, the authors.